



# Oxygenated volatile organic carbon in the western Pacific convective centre: ocean cycling, air-sea gas exchange and atmospheric transport

Cathleen Schlundt[1,2], Christa A. Marandino[1], Susann Tegtmeier[1], Sinikka T. Lennartz[1], Astrid Bracher[3,4], Wee Cheah[3,5], Kirstin Krüger[6], Birgit Quack[1]

[1]GEOMAR Helmholtz Centre for Ocean Research, Kiel, Düsternbrooker Weg, 20, 24105 Kiel, Germany

[2]now at: Josephine Bay Paul Center, Marine Biological Laboratory, Woods Hole, 7 MBL Street, 02540, MA, USA

[3]Alfred Wegener Institute for Polar and Marine Research, Bremerhaven, 27570, Germany

[4]Institute of Environmental Physics, University of Bremen, Bremen, 28359, Germany

[5]now at: Research Center for Environmental Changes, Academia Sinica, Taipei, 128 Academia Road, 11529 Taipei, Taiwan

[6]Meteorology and Oceanography Department of Geosciences, University of Oslo, Oslo, 0315, Norway

*Correspondence to*: Cathleen Schlundt (cschlundt@mbl.edu)

**Abstract.** A suite of oxygenated volatile organic compounds (OVOCs - acetaldehyde, acetone, propanal, butanal and butanone) were measured concurrently in the surface water and atmosphere of the South China Sea and Sulu Sea in November 2011. A strong correlation was observed between all OVOC concentrations in the surface seawater along the entire cruise track, except for acetaldehyde, suggesting similar sources and sinks in the surface ocean. Additionally, several phytoplankton groups, such as haptophytes or pelagophytes, were also correlated to all OVOCs indicating that phytoplankton may be an important source for marine OVOCs in the South China and Sulu Seas. Humic and protein like fluorescent dissolved organic matter (FDOM) components seemed to be additional precursors for butanone and acetaldehyde. The atmospheric OVOC mixing ratios were relative high compared with literature values, suggesting the coastal region of North Borneo as a local hot spot for atmospheric OVOCs. The flux of atmospheric OVOCs was largely into the ocean for all 5 gases, with a few important exceptions near the coast of Borneo. The calculated amount of OVOCs entrained into the ocean seemed to be an important source of OVOCs to the surface ocean. When the fluxes were out of the ocean, marine OVOCs were found to be enough to control the local measured OVOC distribution in the atmosphere. Based on our model calculations, at least 0.4 ppb of marine derived acetone and butanone can reach the upper




troposphere, where they may have an important influence on hydrogen oxide radical formation over the western Pacific Ocean.

## 5    1 Introduction

Oxygenated volatile organic compounds (OVOC) are comprised of ketones, aldehydes and alcohols. They are ubiquitous throughout the troposphere, where they influence the oxidative capacity and air quality. They are involved in the production of reactive nitrogen compounds, such as nitrogen dioxide ($NO_2$, involved in ozone production), peroxynitric acid ($HNO_4$), and nitric acid ($HNO_3$), and they are precursors of peroxyacetyl nitrate

(PAN), a persistent harmful pollutant (Folkins and Chatfield, 2000). OVOCs are a source for hydrogen oxide radicals ($HO_x$), which is of special importance for the upper troposphere (UT), where the concentration of a main precursor, namely water vapor, is much lower than at the Earth surface (Singh et al., 1995;Wennberg et al., 1998;Müller and Brasseur, 1999). Furthermore, OVOCs can contribute to particle formation in the atmosphere, resulting in albedo enhancement (Blando and Turpin, 2000).

The distribution of OVOCs in the atmosphere is determined by a variety of different sources and sinks. Terrestrial emissions from living and decaying plants and oxidation of hydrocarbons in the atmosphere are believed to be the main sources of atmospheric OVOCs. Additionally, biomass burning and anthropogenic emissions play a crucial role for the atmospheric OVOC cycle. Atmospheric sinks of OVOCs include photolysis,

oxidation, and their dry and wet deposition over land and ocean (Heikes et al., 2002;Jacob et al., 2002;Jacob et al., 2005). Based on measurements and model calculations, the strength of atmospheric OVOC sources and sinks are still unbalanced, indicating that unknown global sources and sinks of OVOCs exist (Singh et al., 2003;Jacob et al., 2002). It is believed that the ocean plays a crucial role for atmospheric OVOC concentrations, however it is still poorly understood how the ocean impacts the atmospheric OVOC budget (Heikes et al., 2002;Mincer and

Aicher, 2016;Williams et al., 2004). The main production pathway of OVOCs in the ocean seems to be the photochemical and/or photosensitized oxidation of dissolved organic matter (DOM) in the surface ocean followed by rapid consumption by bacteria (Mopper and Stahovec, 1986;Beale et al., 2013;de Bruyn et al., 2013;de Bruyn et al., 2011;Dixon et al., 2013). The production of OVOCs by specialized bacteria or picophytoplankton seems to be a rather minor source in the ocean (Nemecek-Marshall et al., 1995;Nuccio et al.,

1995;Sunda and Kieber, 1994). Oceanic sinks of OVOCs are their photochemical destruction, air-sea gas exchange or turbulent mixing into the deep ocean (Carpenter *et al.* (2012) and references therein).

It is still debated if the ocean is rather a source or sink for atmospheric OVOCs, given that studies have shown contrasting results. For instance, direct flux measurements of acetone in distinct oceanic regions, showed that the

North Pacific and North Atlantic Ocean were sinks for acetone, while the subtropical Atlantic was observed to be a source, and the South Atlantic was a net zero flux region. Additionally, global flux estimations, when averaged and scaled to the global ocean, showed a wide range between -48 to -1 Tg yr$^{-1}$ (Marandino et al., 2005;Yang et al., 2014a). The ocean appears to be a source for acetaldehyde, with an estimated global flux ranging from 3 to 175 Tg yr$^{-1}$ based on model calculations and direct flux measurements (Beale et al.,

2013;Millet et al., 2010;Singh et al., 2004;Yang et al., 2014a). Model calculations suggest that the Pacific Ocean





is a source for propanal contributing 45 Tg yr$^{-1}$ to the atmosphere (Singh et al., 2003). To the best of our knowledge no butanal or butanone fluxes have been reported to date.

When OVOCs are emitted from the ocean into the marine boundary layer (MBL), they can either accumulate in
the MBL, or be redistributed by deep convection into the mid and upper troposphere throughout the year (Apel et al., 2012). The tropical West Pacific is of special importance because it is an effective entrance region of trace gases even into the upper troposphere and lower stratosphere due to the frequent deep convection (e.g. Aschmann et al., 2009). Even short lived substances, such as DMS or methyl iodide with lifetimes of hours to days, can be entrained into the tropical tropopause layer and can reach the stratosphere (Tegtmeier et al.,
2013;Marandino et al., 2013).

We present the first study of the distribution of a suite of OVOCs (acetaldehyde, acetone, propanal, butanal and butanone) in the surface ocean and atmosphere in the most western region of the Pacific Ocean, the South China Sea and the Sulu Sea, measured in November 2011. In past studies, OVOCs were generally measured either
exclusively in the ocean or in the atmosphere (Dixon et al., 2011;Elias et al., 2011;Singh et al., 2001). Only a few studies have measured acetaldehyde and acetone simultaneously in the water and air (Yang et al., 2014a;Yang et al., 2014b). However, the transport and distribution of the oceanic OVOCs in the upper troposphere was never investigated. We present a comprehensive study of the potential controls on the distribution of OVOCs in the surface ocean, their air-sea flux, and their horizontal and vertical atmospheric
transport over the western Pacific Ocean. Additionally, the possible influence of OVOCs on atmospheric chemistry in the UT is discussed.

## 2 Methods, data analysis and model

### 2.1 Sampling site

During the SHIVA (Stratospheric ozone: Halogen Impacts in a Varying Atmosphere) cruise from Singapore (November 15, 2011) to Manila (November 29, 2011), the German R/V *Sonne* crossed the southern South China Sea, along the northwestern coast of Borneo and entered the Sulu Sea through the Balabac Strait (Figure 1). North easterly trade winds (the median of the wind direction: 50° – 60°), with a mean wind speed of 5.8 ± 2.9
ms$^{-1}$, prevailed during the cruise. The observed mean surface air temperature of 28.3 ± 0.8°C was on average 0.8 ± 0.8°C below the sea surface temperature with a mean of 29.2 ± 0.5°C, benefiting convective activity and precipitation events. The distinct transport of water vapour to the mid troposphere was seen in elevated humidity up to about 6 km, which did not coincide with the marine boundary layer height of 420 ± 120 m, reflecting the characteristics of an unstable, convective well ventilated tropical boundary layer. A detailed overview of the
SHIVA campaign is given by Fuhlbrügge et al. (2016).

### 2.2 OVOC measurements

Oceanic samples (n=90) were collected from day of the year (DOY) 321.9, corresponding to 3.63˚N and 110.34˚E and atmospheric samples (n=37) from DOY 325.3 and 4.6˚N and 113.1˚E (Fig. 1). The samples were
analysed for ethanal (acetaldehyde, CH$_3$CHO), propanone (acetone, (CH$_3$)$_2$CO), propanal (propionaldehyde,





CH$_3$CH$_2$CHO), butanal (butyraldehyde, CH$_3$(CH$_2$)$_2$CHO) and butanone (methyl ethyl keton, CH$_3$C(O)CH$_2$CH$_3$) using a purge and trap system coupled to a gas chromatograph and a mass spectrometer (GC-MS, GC: Agilent Technologies, 7890A; MS: Agilent Technologies, 5975C MS, single quadrupole).

Oceanic samples were taken from an underway pumping system installed in the hydrographic shaft (6 m depth) every three hours. The samples were collected bubble free in 250 ml glass bottles sealed with gas tight Teflon (PTFE) coated lids and were measured immediately after sampling. 10 ml of unfiltered seawater was transferred from the sampling bottle into the purge chamber using a gas tight syringe. OVOCs were expelled from the seawater with a helium flow of 20 ml min$^{-1}$ for 20 min and were trapped in 1/16 inch Sulfinert® stainless steel

tubing submerged in liquid nitrogen. Potassium carbonate (K$_2$CO$_3$) within a 9 cm length glass tube, 0.5 cm in diameter, was used as moisture trap. Hot water was used to transfer the trapped OVOCs on the GC column (fused silica capillary column Supel-QTM Plot, 30m x 0.32 mm) ending in the MS. Water standards were prepared by injecting liquid OVOCs into pure 18 MΩ Milli Q water and were measured in the same way as water samples. The mean analytical errors of the water samples were: acetaldehyde 4.9 %, acetone 20.9 %,

propanal 13.3 %, butanal 12.8 % and butanone 7.8 %. The reproducibility of the system for the water measurements was 20 %. Due to the high solubility of OVOCs in water, it was not possible to expel the gases entirely from the water. Thus, we adjusted all parameters influencing the purge procedure, such as water temperature of the purge chamber (30°C), helium gas flow (20 ml min$^{-1}$) and purging time (20 min), to ensure the reproducibility of the purging procedure between calibrations and samples.

Atmospheric samples were taken in conjunction with water samples from the bow of the ship (about 10 m above seawater surface) by using a portable pump and trap system. Air was pumped for 10 min with a flow of 80 ml min$^{-1}$ through a K$_2$CO$_3$ moisture trap and was concentrated in a Sulfinert® stainless steel tube submerged in liquid nitrogen. We did not sample at the bow of the ship if the relative wind direction was circulating or coming

from aft to avoid contamination of our samples by ship emissions. After trapping, the sample tubing was immediately connected to the GC-MS to avoid loss of sampled compounds. The air samples were transferred to the GC-MS using hot water and were measured in the same way as the water samples. A gas standard mixture of acetaldehyde, acetone, propanal, butanal and butanone in nitrogen (all at mixing ratios of 1 ppm, produced by Apel-Riemer, USA) was trapped and measured in the same way as air samples. The mean analytical errors of the

atmospheric samples were: acetaldehyde 6.3 %, acetone 13.5 %, propanal 14.2 %, butanal 18.7 % and butanone 12.5 %. The reproducibility of the system for the air measurements was 10.5 %.

### 2.3 Phytoplankton pigment analysis

Water samples for phytoplankton pigment analysis were taken from the underway pump system and from Niskin

bottles attached to a rosette equipped with a CTD. The samples were filtered through Whatman GF/F filters (0.7 µm pore size) and were immediately shock frozen in liquid nitrogen and stored at -80°C on board. Pigment extraction and analysis were done in the laboratory at Alfred-Wegener-Institute using the high performance liquid chromatography (HPLC) technique according to the method of Barlow et al. (1997) and modified as described in Taylor et al. (2011). In brief, filtered samples were extracted in 1.5 mL of 100 % acetone plus 50 µL

canthaxanthin as internal standard solution and analyzed by HPLC using a Waters 717plus autosampler, a Waters 600 controller, a LC Microsorb C8 column, and a Waters 2998 photodiode array detector. Part of the





pigment data have been reported in Soppa et al. (2014) and all data are available at the PANGAEA database
(https://doi.pangaea.de/10.1594/PANGAEA.848589).

Based on the pigment concentrations, the corresponding major phytoplankton groups were calculated using the

CHEMTAX software (version 1.95) (Mackey et al., 1996), which employs a 'steepest descent' algorithm to
optimize the marker pigments to chlorophyll a (Chl$a$) ratios identified from HPLC to a given input matrix. Five
input ratio matrices from the Pacific region were chosen as seed values (DiTullio et al., 2003;Higgins and
Mackey, 2000;Zhai et al., 2011;Miki et al., 2008;Mackey et al., 1996). For each seed ratio matrix, sixteen
pigment ratios were randomly generated following the method of the software provider (Wright et al., 2009).

Each of the randomly generated pigment ratios was then used as starting values for CHEMTAX analysis. Six
output matrices with the lowest root mean square error were averaged and used as the "final" starting ratios. A
total of nine phytoplankton groups, namely, diatoms, dinoflagellates, haptophytes, prasinophytes, chlorophytes,
chrysophytes, pelagophytes, prochlorophytes, and cyanobacteria (excluding prochlorophytes) were determined.
Good agreements with significant correlations (Pearson correlation with correlation coefficient $r$) were observed

between five taxa derived from CHEMTAX, and from microscopic and flow cytometry analysis of the same
water samples. The correlations are $r = 0.4$ (prochlorophytes), $r = 0.67$ (*Synechococcus sp.*), $r = 0.75$
(coccolithophores (haptophytes)), $r = 0.93$ (dinoflagellate), and $r = 0.95$ (diatoms).

**2.4 Nutrient measurements**
Dissolved nutrients (phosphate, silicate, nitrate, nitrite) were measured photochemically with a QuAAtro auto-
analyser (SEAL Analytical, UK) according to the method of Grasshoff et al. (1999).

**2.5 Fluorescent dissolved organic matter (FDOM) analysis**

Excitation emission matrices (EEMs) were measured with an F-2700 FL Spectrophotometer (Hitachi) over an
excitation range of 250 to 500nm and an emission range of 280 to 600nm (5nm sampling interval both) using a
1cm quartz cuvette. The slit width was set to 10nm for both. EEMs were blank subtracted, and Raman
normalized, and are reported here in Raman units (RU). A parallel factor analysis (PARAFAC) was performed
for 187 EEMs using the drEEM Toolbox (Murphy et al., 2013) for MATLAB. Six components were split half

validated using alternating initialization.

**2.6 Statistics**
We applied principal component analysis (PCA) to examine how a combination of different physical, chemical

and biological variables, such as the phytoplankton community, nutrient availability or physical parameters
(temperature, salinity), might influence the OVOC concentration and distribution in the surface seawater of the
South China Sea and Sulu Sea. We calculated PCA to reveal a simplified underlying structure of our multivariate
data set and to find the best model explaining the variance of our data. PCA converts a large number of variables
$n$ into a smaller number of artificial variables $q$ (principal components, PC) ($n > q$) with a minimum loss of

information. Prior to the PCA, we normalized our dataset to compare data with different units. Each PC has an
associated eigenvalue, which indicates the variation of the data. The higher the eigenvalue the better the




variation of all the data is explained by the appropriate PC. Furthermore, the factor loadings of the PCs were calculated, which explain the variance of each variable by the corresponding PC. High factor loadings refer to high and significant correlations between variables. In addition to the PCA, we applied the Spearman rank correlation analysis (with the correlation coefficient $r_s$) to find possible direct links between the different OVOCs.

**2.7 Flux calculations**

The oceanic and atmospheric OVOC data were used for flux calculations. Wind speed and sea surface temperature obtained from ship sensors at ten minute resolution were selected for time and positions of OVOC measurements. The flux (F) was calculated according to Johnson (2010):

$$F = -K_a(C_a - K_H C_w) \, , \tag{1}$$

$K_a$ is the total transfer velocity from the gas phase point of view that is composed of the water side single-phase transfer velocity ($k_w$) and the air side single phase transfer velocity ($k_a$) (Johnson, 2010). The 10 m height wind speed dependent $k_w$ determined by Nightingale et al. (2000) was adjusted with the temperature dependent Schmidt number for $CO_2$ that we corrected with the molar volume of each OVOC, respectively, according to Hayduk and Laudie (1974). We used the $k_a$ determined by Duce et al. (1991), which depends on the wind speed in 10 m height and on the molecular weight of the trace gas.

$C_a$ and $C_w$ are the concentrations of OVOCs in the atmosphere (around 10 m above sea level) and in the sea surface water (6 m depth), respectively. $K_H$ is the dimensionless, temperature dependent Henry's law constant, which was described in Sander (1999) and that was modified for each OVOC by using empirically determined apparent partition coefficients for the different OVOCs from Zhou and Mopper (1990).

**2.8 OVOC atmospheric transport modelling**

The atmospheric transport of the OVOCs from the oceanic surface into the MBL was simulated with the Lagrangian particle dispersion model FLEXPART (Stohl et al., 2005). FLEXPART is an off-line model driven by external meteorological fields and includes parameterizations of moist convection, turbulence in the boundary layer, dry deposition, scavenging, and the simulation of chemical decay. This model has been validated extensively with measurements from large scale tracer experiments and has been used in many studies of long range and mesoscale transport (Stohl et al. (2005) and references therein).

For the simulations of atmospheric transport and chemical decay of the emitted oceanic OVOCs, we calculated trajectories of a multitude of air parcels. For each data point of the observed sea to air flux, 10000 air parcels were released from a 0.1° x 0.1° grid box at the ocean surface centered at the measurement location. The air parcels were released over a time period ranging between 1 and 18 days depending on the chemical lifetime of the respective gas as given below. The FLEXPART v9.2 runs were driven by the ECMWF reanalysis ERA-Interim (Dee et al., 2011) given at a horizontal resolution of 1° x 1° on 60 model levels. FLEXPART calculates transport, dispersion and convection of the air parcels from the horizontal and vertical wind fields, temperature, specific humidity, convective and large scale precipitation. The chemical decay of the OVOCs was prescribed by



their atmospheric lifetime, which was set to 14 hours for butanal (Calvert, 2011), 15 hours for propanal (Rosado-Reyes and Francisco, 2007), one day for acetaldehyde (Millet et al., 2010), 10 days for butanone (Calvert, 2011) and 18 days for acetone (Khan et al., 2015). A second set of simulations was conducted to analyze possible sources of the observed atmospheric mixing ratios. While the importance of the emissions was investigated by

forward runs as described above, possible sources of the observed mixing ratios were identified by backward trajectory runs. At each measurement location, 100 trajectories were released at the time of the observation and calculated backward in time over a 24 hour period.

**3. Results and discussion**

**3.1 OVOCs in the surface ocean**

For the first time, a suite of OVOCs were measured in the surface water of the southern part of the South China Sea and adjacent Sulu Sea and in the overlaying marine boundary layer. Over the entire cruise track, average surface seawater concentrations (6 m depth) for acetaldehyde, acetone, propanal, butanal and butanone were 4.1, 21.3, 1, 0.7 and 0.9 nmol L$^{-1}$, respectively (Table 1). Acetaldehyde concentrations were similar to previous

measurements in the surface waters of the open Atlantic and Pacific Oceans (Beale et al., 2013;Kameyama et al., 2010;Mopper and Stahovec, 1986;Yang et al., 2014a;Zhou and Mopper, 1997), and in the lower range compared to coastal concentrations in the English Channel (Beale et al., 2015) (Table 2). In contrast, the concentration of acetone was in the higher range compared to literature values of the open ocean and coastal regions (Beale et al., 2013;Beale et al., 2015;Dixon et al., 2014;Kameyama et al., 2010;Marandino et al., 2005;Williams et al.,

2004;Yang et al., 2014a;Yang et al., 2014b;Zhou and Mopper, 1997). Only a few studies measured propanal, butanal and butanone in the ocean. The concentrations in our study were elevated compared to a study in the open ocean near the Bahamas by Zhou and Mopper (1997) and low compared to a study in the South-East Florida coast (Mopper and Stahovec, 1986) (Table 2). Corwin (1969) measured the same suite of OVOCs as in our study in the Straits of Florida and in the Eastern Mediterranean. Their concentrations were one to three

orders of magnitude higher compared to our values and to literature values listed in Table 2.

The distribution pattern of the OVOCs was variable in the shallow ocean along the coast off Borneo in the South China Sea (Fig. 2 a, b, DOY 323 – 329, <117° E) and showed mainly low concentrations in the open ocean of the Sulu Sea and in the coastal region off the eastern Philippines (Fig. 2 a, b, DOY 329 – 333, >117° E). Slightly

elevated OVOC concentrations occurred close to the coast off Kuching (Fig. 2 a, b, DOY 322 – 323) in conjunction with elevated nitrate and TChl*a* concentrations (Fig. 1 and 2 d). The rivers Sadong, Lupar and Saribas form a large river delta to the east of Kuching that impacts the coastal region. This was visible in a decrease of salinity and slightly elevated nutrient concentrations indicated by nitrate in Fig. 2d due to river outflow that consequently induced a phytoplankton bloom. Elevated OVOC concentrations occurred also around

5° N and 114° E (Fig. 2 a, b, DOY 325) and at the northern tip of Borneo (Fig. 2 a, b, DOY 328 – 329, between 7-8° N and 117-119° E). These two regions were less influenced by river outflow, as indicated by elevated salinity and lower nutrient concentrations compared to the region west of 114° E.

We correlated the OVOCs to each other to find possible relationships between them. Significant correlations

were found between all OVOCs (up to $r_s$ = 0.8, p-value 0.001). However, the coefficient of determination was





relatively low ($r_s = 0.45 - 0.55$, p-value 0.001) for correlations between acetaldehyde and the other OVOCs. Additionally, we examined the relationships between the OVOCs for the entire cruise track using PCA. 75 % of the OVOC variability was explained by the first component of the PCA (Fig. 3a), which points to their similar distribution pattern and their close relationship. When acetaldehyde was excluded from the PCA model, 89 % of

the variability of the remaining OVOCs was explained by the first component. It seems that acetaldehyde was controlled by additional factors than the other OVOCs in the South China Sea and Sulu Sea. Several studies investigated the biogeochemical pathways of acetaldehyde and acetone in different oceanic regions. For instance, de Bruyn et al. (2011) observed higher production rates for acetaldehyde compared with acetone from photochemical oxidation of colored dissolved organic matter (CDOM) in a coastal region of southern California,

USA. In contrast, Dixon et al. (2013) estimated a higher photochemical production rate for acetone compared with acetaldehyde and showed that up to 13 % of acetone and up to 100 % of acetaldehyde were lost in the surface water of the Atlantic Ocean due to microbial oxidation. Furthermore, they estimated a biological lifetime for acetone and acetaldehyde between 5 and 80 days and between 2 hours and 1 day, respectively. Dixon et al. (2014) confirmed that bacteria preferred acetaldehyde rather than acetone as carbon or energy sources in the

English Channel. Mopper and Stahovec (1986) observed a stronger diurnal cycle with highest concentrations in the late afternoon for acetaldehyde compared with acetone. Additionally, Beale et al. (2013) suggested that regions of the Atlantic Ocean might be undersaturated in acetaldehyde, but not in acetone. It seems that acetaldehyde is subject to faster turnover compared to acetone. This might explain the weaker link between acetaldehyde and acetone, and probably also with the other OVOCs, in this study.

### 3.1.1 OVOC concentrations influenced by phytoplankton and FDOM

We applied PCA to understand which environmental factors (nutrients, phytoplankton pigments, biomasses of phytoplankton groups, phytoplankton cell size, salinity, temperature, wind speed, halogenated compounds, different FDOM components, and methane) might be related to or influence the OVOCs in the sampling site. We

found a link between the OVOCs and several phytoplankton groups, including haptophytes, pelagophytes, dinoflagellates, prasinophytes and chlorophytes (Fig. 3 b), all of which made up 37 % biomass on average over the cruise. 53 % of the variability of the OVOCs together with these phytoplankton groups was accounted for by the first component of the PCA. The link between the OVOCs and phytoplankton can either refer to the direct production of OVOCs by phytoplankton or to similar environmental conditions causing the same variability for

both. However, it is unlikely that over the entire cruise track, in both coastal and open ocean regions, the same environmental conditions prevailed that triggered the same OVOC and phytoplankton distribution pattern without any direct connection between the two. We concluded, based on the PCA results, it is possible that phytoplankton were a source of OVOCs in the South China Sea and Sulu Sea.

To the best of our knowledge, no study described the direct production of OVOCs by phytoplankton. Whelan et al. (1982) tested macroalgae and phytoplankton for OVOC production and measured high concentrations of OVOCs (acetone, propanal, butanal, and butanone) in macroalgae, but not in phytoplankton such as diatoms, dinoflagellate, haptophytes or cyanobacteria, therefore, excluding phytoplankton as OVOC producers. Additionally, Beale et al. (2013) found no correlation between acetone production and primary production in the

Atlantic Ocean and, thus, excluded marine phytoplankton as a significant source for acetone. Nuccio et al. (1995) tested axenic monocultures of cyanobacteria and dinoflagellates for their ability to produce acetaldehyde





and propanal. They observed mainly the production of formaldehyde in these cultures. However, a filtered (0.2 µm pore size) seawater sample with a natural assemblage of bacteria- and picoplankton showed an increase of propanal concentration together with Chl*a* concentration, assuming propanal production by picophytoplankton or their symbiotic bacteria. In addition, Sinha et al. (2007) observed significant correlations of the haptophyte

*Emiliania huxleyi* and picophytoplankton with acetone and acetaldehyde emissions in a mesocosm study in Norway, suggesting a possible production of OVOCs by phytoplankton. Mincer and Aicher (2016) showed that a broad range of marine phytoplankton species of different groups produce significant amounts of methanol, the most abundant OVOC in the atmosphere. Comprehensive studies are missing that test a wide range of phytoplankton species for their potential to produce different OVOCs.

If the OVOCs in this study were not produced by phytoplankton, it is possible that they were produced by attached or symbiotic bacteria of the phytoplankton. Several studies showed that marine bacteria can produce acetone, acetaldehyde and, most likely, propanal (Nemecek-Marshall et al., 1995;Nuccio et al., 1995;Sunda and Kieber, 1994;Nemecek-Marshall et al., 1999). However, measurements of bacteria cell abundance and

composition are not available for our study. Thus, incubation experiments have to be conducted in the future to test directly the microbial production and consumption of OVOC's in the ocean.

An additional main source of OVOCs in surface seawater is the DOM pool (Mopper and Stahovec, 1986;Zhou and Mopper, 1997). DOM is composed of non-colored and colored DOM (CDOM) that can be further

subdivided into a fluorescent part of the CDOM pool called FDOM. We identified 6 FDOM components in our study, with 4 humic like substances (component C2-C4, C6), which were composed of refractory humic like compounds that originated from coastal and terrestrial environments, which were either soil derived, segregated by marine algae or altered by microbial activities (C2 described in Tanaka et al. (2014) as C1 and in Yu et al. (2015) as C5; C3 described in Li et al. (2015) as C4; C4 described in Kowalczuk et al. (2009) as C2; C6

described in Kowalczuk et al. (2013) as C5). Furthermore, 2 protein like substances (C1 and C5) were identified, originating from marine environments associated with recent biological production in surface seawater (C1 described in Kowalczuk et al. (2013) as C3; C5 described in Wünsch et al. (2015) as C1). For a detailed overview of the FDOM distribution patterns, see the supplemental material (S1).

Cell fragments or compounds excreted by phytoplankton are important parts of the DOM pool, including FDOM, and can be potential precursors of OVOCs when they undergo photolysis (de Bruyn et al., 2011). Thus, a close link between FDOM and OVOCs could additionally explain the observed relationship between phytoplankton and OVOCs. We conducted PCA including OVOCs and the six different FDOM components and found a link between the components C2-C5 with acetaldehyde and butanone along the coast of Borneo (PC1

60.7 %, Fig. 3 c). When only data of the open ocean of the Sulu Sea was considered for PCA no relationship was found, thus, only coastal derived FDOM seemed to influence the acetaldehyde and butanone. The link between acetaldehyde and the FDOM components is in line with the findings of other studies investigating CDOM, including FDOM (de Bruyn et al., 2011;Dixon et al., 2013;Kieber et al., 1990). However, Dixon et al. (2013) observed higher photochemical production of acetone compared to acetaldehyde, suggesting CDOM as the main

source for acetone. de Bruyn et al. (2011) emphasized that the production of OVOCs can vary significantly and regionally depending on the CDOM source. The relationship between acetaldehyde and FDOM supports our





findings that acetaldehyde is controlled by additional parameters compared to the other OVOCs of this study. Additionally, butanone seemed to be controlled by FDOM derived from marine phytoplankton, because of its close link to both phytoplankton and FDOM.

### 3.1.2 Atmospheric OVOCs as a source for OVOCs in surface water

We calculated the air-sea gas exchange of OVOCs between the ocean and the atmosphere to understand if the ocean is either a source or a sink for atmospheric OVOCs. On average, we observed a flux into the ocean (for details see section 3.2 below) suggesting that atmospheric OVOCs are a source for oceanic OVOCs. We

calculated the percentage contribution of atmospheric OVOCs to the marine OVOC pool in the surface ocean. We assumed that all sources (S) equal all sinks (L) in the ocean and that the sinks of OVOCs can be determined by the OVOC concentrations divided by their lifetimes ($\tau$):

$$S = L = \frac{[ovoc]}{\tau}, \tag{2}$$

Dixon et al. (2013) determined a lifetime for acetaldehyde between 2-5 hours and for acetone between 5-55 days in open oceanic environments. To be conservative, we used for our calculation the lower range of 2 hours up to 5 days. To the best of our knowledge, no lifetimes were determined for butanone, butanal and propanal in the ocean, thus, we assume their lifetime in the same range as acetone and acetaldehyde. We determined the OVOC

concentration for the entire mixed layer, which was on average 37 m deep along our cruise track. With an OVOC lifetime of 2 hours, the atmospheric OVOC contribution to the marine pool is of minor importance (0.3 – 0.7 %). The contribution increases to 5 – 9 % when 1 day lifetime is assumed and up to 20 – 44 % when a lifetime of 5 days is assumed. Based on these results, atmospheric OVOCs can be an important source for the OVOC pool in the surface ocean in the South China and Sulu Seas.

### 3.2 OVOCs in the atmosphere

Over the entire cruise track, average atmospheric mixing ratios (10 m above sea level) for acetaldehyde, acetone, propanal, butanal and butanone were 0.86, 2.1, 0.15, 0.06 and 0.06 ppb, respectively (Table 1, Fig. 4). The

values of acetaldehyde and butanone were on average double the values found in previous studies (compare Table 1 and 3). Acetone and propanal were even one order of magnitude higher than literature values. Similar elevated atmospheric acetaldehyde mixing ratios were measured at the west coast of Ireland (Mace Head Observatory). Wisthaler et al. (2002) measured similar levels for acetaldehyde and acetone in continental air masses they encountered above the Indian Ocean. In comparison to most studies listed in Table 3, the coastal

region of Borneo seems to be a regional hot spot for atmospheric OVOCs, which might originate from anthropogenic activities, from terrestrial vegetation and/or due to gas exchange between the ocean and atmosphere.

Based on our calculated fluxes, over much of the cruise track, the ocean and atmosphere appeared to be near

equilibrium for all compounds (Fig. 5). Nonetheless, the average computed fluxes in the South China Sea and



Sulu Seas were negative (into the ocean), except for butanal (Table 1). The on average negative values are caused by localized, strong sinks such as observed in the Balabac Strait (Fig 5, a-e, DOY 328) and the open ocean of the Sulu Sea (DOY >330). In contrast, along the Borneo coast, the ocean was mainly a source for all OVOCs, except for acetaldehyde (Fig. 5, DOY 325-328). Observations in the open Atlantic and Pacific Ocean

found acetaldehyde and propanal fluxes out of the ocean, in contrast to our findings (Tables 1 and 3; Yang et al. (2014a), Zhou and Mopper (1993) and Singh et al. (2003)). Our results are in agreement with previous studies showing that acetone is transported from the atmosphere into the ocean (Table 3). To the best of our knowledge no fluxes were ever reported for butanal and butanone.

The computed fluxes from ocean to atmosphere around the Borneo coast were unexpected, due to the close proximity of the cruise track to the coast, the known continentally based sources to the atmosphere, and the high values of atmospheric OVOCs that we measured. These fluxes from ocean to atmosphere imply that the production of OVOCs in the ocean around the Borneo coast must be large in order to overcome the high atmospheric mixing ratios observed there. Therefore, this region is a potentially important local source of

OVOCs to the atmosphere. It is of interest to know if the calculated ocean atmosphere fluxes are enough to explain the observed atmospheric OVOC mixing ratios measured along the cruise track or if these must result from other sources. In addition, we investigated the transport of OVOCs from ocean emissions to the UT, where they can influence $HO_x$ formation.

### 3.2.1 Horizontal OVOC transport
For all measurement locations with a positive flux from the ocean to the atmosphere, we use FLEXPART simulations (see Section 2.6) to derive the OVOC mixing ratios within the MBL. We release trajectories continuously over the whole lifetime of the respective gas loaded with the amount of the OVOC prescribed by

the observed emissions at this location. Based on the FLEXPART simulations, we derived the mean OVOC mixing ratios from all air parcels released from one measurement location. Given the low density of existing measurements over the area of interest, we do not take into account mixing with air parcels impacted by other (unknown) source regions. This assumption is relatively realistic for the shorter lived OVOCs with lifetimes of a few hours (butanal and propanal), but less realistic for OVOCs with longer lifetimes of days to weeks (butanone

and acetone). For the latter, transport and mixing will have a noticeable impact on the actual mixing ratios. Ignoring the transport and mixing processes, as done by our method, is equivalent to the assumption that emissions are constant over time (lifetime of the respective OVOC) and space (region specific by transport time scales equivalent to the respective life time). Under these assumptions our method offers a useful concept to estimate the maximum atmospheric mixing ratios that could result from the observed oceanic sources.


Figure 6 shows the simulated (left panel) and observed (right panel) mixing ratios of butanone (upper panel) and propanal (lower panel) depicted at the position of the corresponding measurements. All positions where the flux was from the ocean into the atmosphere are marked by a black edge around the symbol. FLEXPART mixing ratios for butanal based on the observed sea to air fluxes were on average 0.033 ppb with maximum values of

0.11 ppb. These values agree very well with the observed mixing ratios of butanal west of 116° E with a mean of 0.039 ppb and a maximum value of 0.16 ppb. However, east of 116° E, the observed butanal mixing ratios were





on average 0.21 ppb and were thus more than 6 times larger than what could be explained by the local ocean sources. In particular, observed maximum values of up to 1.2 ppb must be driven by other nearby sources. For propanal, positive fluxes occur along the coast off Borneo in the South China Sea but not at the northern tip of Borneo, with one exception around 120° E. Atmospheric mixing ratios based on these fluxes are around 0.017

ppb, which is smaller than the mixing ratios observed west of 116° E with a mean of 0.037 ppb. One can conclude that the local oceanic sources can on average explain about half of the observed mixing ratios in this region. East of 116° E, however, the observations are again much larger (0.36 ppb) and cannot be explained at all by local oceanic sources.

Butanone and acetone have longer lifetimes and released air parcels will spread over a larger area within the respective lifetimes. Thus we do not show the simulated mixing ratios at the position of the measurements as it was done for butanal and propanal, but instead compared the mean mixing ratios from observations and model simulations (Fig. 7). The observed atmospheric mixing ratios were split into two regimes west and east of 116° E, with relatively low values to the west and higher values to the east for both compounds. Also for these two

longer-lived OVOCs, the observed atmospheric mixing ratios west of 116° E were very similar to the maximum amount that can be derived from the observed oceanic sources (Figure 7). For butanone, observation of 0.07 ppb agreed very well with model results of 0.06 ppb. The acetone mixing ratios were much higher, again with a very good agreement between the model results (0.65 ppb) and observations (0.55 ppb). East of 116° E, however, the observed mixing ratios were again much larger (2.1 ppb for butanone and 4.4 ppb for acetone) than what could

be explained by oceanic sources, consistent with what has been found for the shorter lived OVOCs. Overall, the observed atmospheric OVOCs can be split into two regimes, where one can be explained by the oceanic sources and the other requires additional terrestrial or anthropogenic sources.

To further explore the higher mixing ratios observed east of 116° E, we focused on the shorter lived OVOCs,
butanal and propanal, using the backward trajectory calculations over one day. The backward trajectories (Figure 8) display air mass transport from east to west over the 24 hours preceding each measurement. For better visibility, we split the trajectories into five different groups according to slightly different transport patterns. Groups 1, 2 and 3 (green, light blue and magenta trajectories), encompassing most air masses observed east of 116°E, crossed the coastline of the Philippines within this time period. Most air masses west of 116° E (red and

blue trajectories), however, were too far from the Philippine coast to have experienced terrestrial or anthropogenic influence in the 24 hours before these samples were taken onboard of the ship. The inability of the oceanic sources to explain the high atmospheric mixing ratios east of 116° E strongly suggests additional terrestrial or anthropogenic sources. The backward trajectory analysis further supports this hypothesis.


### 3.2.2 Implications of oceanic sources on the UT

To assess the potential importance of this local OVOC source for upper tropospheric $HO_x$ formation, we computed the vertical transport using FLEXPART for acetone, butanone, propanal, and butanal (Figure 9). Acetaldehyde was left out of this analysis, because there were only 2 data points with fluxes out of the sea
surface to the atmosphere. The vertical trajectory mixing ratios were calculated by emitting the average value of the computed sea to air fluxes over one month in the source location and prescribing tropospheric lifetimes for





the loss term (see Methods). For the aldehydes, the highest mixing ratios were computed for the lower troposphere, with values up to approximately 0.01 ppb. Values aloft, up to approximately 15 km, did not exceed 0.003 ppb for both propanal and butanal. The ketone vertical profiles showed maxima in both the lower 2 km and above 10 km, with values exceeding 0.30 ppb for acetone and 0.10 ppb for butanone. This is likely due to

the longer tropospheric lifetimes of the ketones than the aldehydes, as the computed ketone sources to the atmosphere were not always larger than the aldehydes (Figure 5).

It was previously thought that low levels of water vapor in the UT prevent $HO_x$ formation there. In the 1990s, it had become clear that this idea was not valid (Singh et al., 1995;Chatfield and Crutzen, 1984;Prather and Jacob,

1997). Wennberg et al. (1998) sought to understand observed levels of OH in the troposphere by using both the O(1D) + $H_2O$ reaction, as well as acetone photolysis (using approximately 0.30 ppb of acetone). They found that at heights between 9 and 14 km, the inclusion of acetone photolysis improved the model measurement agreement. At heights from 14-16 km, the inclusion of acetone photolysis resulted in near perfect agreement between model and measurement, suggesting acetone as the main source for OH at these heights. However, in

2004, Blitz *et al.* revised the acetone quantum yield downward, resulting in less acetone loss, which would also result in lower values of computed OH. It is likely that other OVOCs with similar photochemical properties, such as butanone, are also present in the UT and could compensate this downward revision. Here we calculate that a combined value greater than 0.4 ppb of acetone and butanone from a spatially limited ocean source region in the South China Sea can be present in the UT (Figure 9). Data recently obtained from the CARIBIC campaign

show that acetone values between 0.30 and 1.2 ppb are in fact observed in the UT (Neumaier et al., 2014). By combining their data with EMAC chemistry simulations, Neumaier et al. (2014) show that UT $HO_x$ production from acetone is significant all year round, reaching an average of 60 % of that from ozone photolysis in summer and 95 % in fall. It should be noted that CARIBIC took place in the midlatitudes, not in the tropics as SHIVA, but their reported $HO_x$ precursors were similar to SHIVA (water vapor range: CARIBIC 20-140 ppm, SHIVA 0-

40 ppm; ozone: CARIBIC 50-150 ppb, SHIVA 20-100 ppb (data not shown)). Thus, ketones in the UT from ocean sources in the western Pacific Ocean have the potential to contribute at least 30 % of UT mixing ratios, having an important influence on $HO_x$ formation there. In addition, Neumaier et al. (2014) found, using sensitivity studies with EMAC, that one of the major uncertainties in quantifying UT $HO_x$ formation from acetone is the acetone mixing ratio distribution. Therefore, more observations and forward trajectory calculations

from the atmospheric boundary layer are needed to understand the important role of acetone and other longer lived OVOCs on the distribution of $HO_x$ in the UT.

**4. Summary**

For the first time, a suite of OVOCs were measured simultaneously in the surface water and overlaying atmosphere in the South China Sea and Sulu Sea. Sea surface concentrations of acetone, propanal, butanal and butanone correlated with each other, indicating similar sources and sinks in the surface water. Phytoplankton seemed to be the main source for acetone, propanal and butanal in the South China and Sulu Sea. Acetaldehyde and butanone seemed to be produced by both phytoplankton and terrestrial and marine derived FDOM

components.



The South China Sea seemed to be a regional hot spot for atmospheric OVOCs and the air-sea gas exchange was on average into the ocean, thus, atmospheric OVOCs seemed to be an additional important source (up to 44 %) for OVOCs in the surface ocean. However, local fluxes from the ocean into the atmosphere along the coast off Borneo, implied that local coastal marine sources can be sufficient to drive fluxes from ocean to atmosphere, even when atmospheric mixing ratios were large. West of 116° E the flux of marine OVOCs into the atmosphere could explain the atmospheric mixing ratios of OVOCs, while east of 116° E terrestrial and anthropogenic sources were responsible for the elevated atmospheric OVOCs. The longer lived marine derived ketones, acetone and butanone, were calculated to contribute more than 0.4 ppb to the UT in this convective region and may be important for $HO_x$ formation above the South China Sea.

**Data availability**

All data (seawater and atmospheric OVOC data, OVOC flux data, nitrate, salinity, and FDOM) can be retrieved from the supplemental material and will be available at the PANGAEA database (https://doi.pangaea.de/10.1594/PANGAEA.848589). Phytoplankton data are already available at the PANGAEA database.

**Author contribution**

C. Schlundt and C. Marandino designed the experiments and measured the samples of OVOCs and FDOM. C. Schlundt performed statistical calculations, wrote most of the sections 1-3.2 first paragraph and summary, and created Fig. 2-5 and Fig. S1. C. Marandino wrote sections 3.2 and 3.2.2. S. Tegtmeier performed the Flexpart model analysis, wrote 2.8 and 3.2.1 and created Fig. 6-9. S. Lennartz performed the FDOM PARAFAC calculations and wrote section 2.5. W. Cheah took phytoplankton samples, measured them, performed the CHEMTAX analysis and wrote together with A. Bracher section 2.3. A. Bracher created Fig. 1. B. Quack wrote section 2.1. All authors contributed to review and improve the text.

**Acknowledgements**

Thanks to the captain and crew of the R/V *Sonne*. We thank Sonja Wiegmann, Mariana Soppa and Joseph Palermo supporting the phytoplankton sampling during the cruise and Sonja Wiegmann for the HPLC pigment analysis. We gratefully acknowledge the NASA Goddard Space Flight Center, Ocean Biology Processing Group for providing SeaWiFS Ocean Color Data CDOM. We thank Francois Steinmetz (HYGEOS) for supplying Polymer-MERIS CHL data (also for campaign planning) and ESA for MERIS level-1 satellite data. This work was supported by the EU project SHIVA under grant agreement no. FP7-ENV-2007-1-226224 and by the BMBF grants SHIVA-*Sonne* (FKZ: 03G0218A). We gratefully acknowledge the "Studienstiftung des Deutschen Volkes" for providing the "Promotionsstipendium" for Cathleen Schlundt to conduct this research study. Astrid Bracher's contribution was also partly funded by ESRIN/ESA within the SEOM (Sceintific Exploration of operational missions) - Sentinel for Science Synergy (SY-4Sci Synergy) program via the project SynSenPFT. Additional funding for C.S, C.A.M., and S.T.L. came from the Helmholtz Young Investigator Group of C.A.M.,





TRASE-EC (VH-NG-819), from the Helmholtz Association through the President's Initiative and Networking Fund and the GEOMAR Helmholtz-Zentrum für Ozeanforschung Kiel.

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





Table 1: Averages and ranges of water and air concentrations of OVOCs and their air-sea gas exchange rates. Negative flux values refer to gas exchange from the atmosphere into the ocean.

|  | Water | | Air | | Flux | |
|---|---|---|---|---|---|---|
|  | Median | Range | Median | Range | Median | Range |
|  | [nmol L$^{-1}$] | | [ppb] | | [μmol m$^{-2}$ d$^{-1}$] | |
| Acetaldehyde | 4.11 | 0.35 - 14.45 | 0.86 | 0.11 - 8.5 | -10.11 | -139.36 – 7.27 |
| Acetone | 21.33 | 2.47 - 67.76 | 2.1 | 0.14 - 14.48 | -18.34 | -166.14 – 30.63 |
| Propanal | 1.04 | 0.08 - 3.29 | 0.15 | 0.01 - 1.42 | 0.03 | -7.49 – 2.58 |
| Butanal | 0.71 | 0 - 3.35 | 0.06 | 0.006 - 1.21 | 0.5 | -6.56 – 2.99 |
| Butanone | 0.88 | 0.11 - 4.31 | 0.06 | 0.003 - 35.46 | -0.71 | -344.82 – 1.96 |

Table 2: Literature values of OVOCs in the open and coastal oceans.

|  | Open Ocean | Mean [nmol L$^{-1}$] | Range | Date month year | Study |
|---|---|---|---|---|---|
| Acet-alde-hyde | Atlantic Ocean |  | 3 - 9 | 10.-12.2009 | Beale *et al.* 2013 |
|  | North Pacific Ocean |  | BLD - 5.9 | 7.-8. 2008 | Kameyama *et al.* 2010 |
|  | 100 km East of Bahamas | 1.38 ±0.08 |  | 3.1989 | Zhou and Mopper 1997 |
|  | South-west Coast of Florida |  | 2 - 30 | 4.1985 | Mopper and Stahovec 1986 |
|  | Atlantic Ocean | ~6 | 3 - 9 | 10.-11.2012 | Yang *et al.* 2014 a |
| Acetone | Atlantic Ocean | 8 | 2 - 24 | 10.-12.2009 | Beale *et al.* 2013 |
|  | North Pacific Ocean | 18.9 | 4.4 - 41.3 | 7.-8.2008 | Kameyama *et al.* 2010 |
|  | Tropical Atlantic Ocean | 17.6 |  | 10.-11.2002 | Williams *et al.* 2004 |
|  | 100 km East of Bahamas | 3 ±-0.23 |  | 3.1989 | Zhou and Mopper 1997 |
|  | Western Tropical Pacific |  | 1.8 - 27.2 | 5.-7.2004 | Marandino *et al.* 2005 |
|  | Western English Channel |  | 3 - 7.5 | 2.-6.2011 | Dixon *et al.* 2014 |
|  | Atlantic Ocean | 13.7 | 4 - 36 | 10.-11.2012 | Yang *et al.* 2014 b |
|  | North Atlantic | 5.7 | 3 - 9 | 10.-11.2013 | Yang *et al.* 2014 a |
|  | Eastern Mediterranean |  | 310 – 912 | 8.1965 | Corwin *et al.* 1969 |
| Propa-nal | 100 km East of Bahamas | 0.4 +/-0.06 |  | 3.1989 | Zhou and Mopper, 1997 |
| Butanal | 100 km East of Bahamas | 0.2 +/-0.06 |  | 3.1989 | Zhou and Mopper, 1997 |
| Buta-none | 100 km East of Bahamas | < 0.5 |  | 3.1989 | Zhou and Mopper, 1997 |
|  | Eastern Mediterranean |  | 69 – 111 | 8.1965 | Corwin *et al.* 1969 |



|  | Coastal Ocean | Mean [nmol L$^{-1}$] | Range | Date month year | Study |
|---|---|---|---|---|---|
| Acetald. | Western English Channel |  | 4 - 37 | all year 2011/12 | Beale *et al.* 2015 |
| Acetone | Western English Channel |  | 2 - 10 | 2011/12 | Beale *et al.* 2015 |
| Propa-nal | Straits of Florida Biscayne Bay, South East Florida | 4 | 241 - 895 | February 1968 Feb. 1986 | Corwin *et al.* 1969 Mopper and Stahovec, 1986 |
| Butanal | Strait of Florida |  | 180 - 666 | February 1968 | Corwin *et al.* 1969 |
| Buta-none | Straits of Florida |  | 111 - 305 | February 1968 | Corwin *et al.* 1969 |

Table 3: Literature values of OVOC concentrations in the atmosphere and their air-sea gas exchange.

|  | Region | Height [km] | Mean [ppb] | Range [ppb] | Flux[a] [μmol m$^{-2}$ d$^{-2}$] | Date | Study |
|---|---|---|---|---|---|---|---|
| Acetal-dehyde | Cape Verde (Atlantic Ocean) | 0.01 | 0.43 ±0.19 | 0.19 - 0.67 |  | annual 2006 - 2011 | Read *et al.* 2012 |
|  | Pacific Ocean 40°N - 10°S, 125°W - 140°E | 0.005 – 0.01 |  | 0.03 - 0.1 |  | 2.1994 | Singh *et al.* 1995 |
|  | Pacific Ocean (10-45°N, 100-230°E) | 0 -2 | 0.2 ±0.04 |  | 0.002 | Winter/ spring 2011 | Singh *et al.* 2003 |
|  | Pacific Ocean (10-45°N, 100-230°E) | 2- 4 | 0.17 ±0.05 |  | 0.002 | Winter/ spring 2011 | Singh *et al.* 2003 |
|  | North Atlan. O., Mace Head observatory | 0.025 | 0.44 | 0.12 - 2.12 |  | 7.-9 2002 | Lewis *et al.* 2005 |
|  | Pacific Ocean (10-50 °N) | 0.1-12 | 0.12 ±0.06 |  |  | 2.-4.2001 | Singh *et al.* 2004 |
|  | Atlantic Ocean (50°N - 50°S, 10-60°W) | up to 12 |  | 0.06 - 0.1 |  | 3.-4.1999 | Sing *et al.* 2001 |
|  | Trop. west. Atl. O. (10- 30°N, 60 - 80°W) | 0.018 |  | > 0.05 - 0.25 | 0.6 ±2.5[b] | 10.-11. 2012 | Yang *et al.* 2014 (a) |
|  | North Atlantic, Mace Head | 0.025 |  | 0.11 – 0.24 |  | 10.1993 – 4.1995 | Solberg *et al.* 1996 |
|  | Indian Ocean | 0.028 | 0.21 ±0.3 | 0.12 – 0.5 |  | 3.1999 | Wisthaler *et al.* 2002 |
|  | Caribbean Sea | 0.01 | 0.57 ±0.3 | 0.2 – 1.4 |  | 10.1988 | Zhou and Mopper, 1993/1997 |
|  | 100 km East of Bahamas |  |  |  | 17.28 | 3.1989 | Zhou and Mopper, 1997 |





| | | | | | | |
|---|---|---|---|---|---|---|
| Acetone | Cape Verde, Atlantic Ocean | 0.01 | 0.55 | 0.23 - 0.91 | | annual 2006 - 2011 | Read *et al.* 2012 |
| | Pacific Ocean 40°N - 10°S, 125°W - 140°E | 0.005 – 0.01 | | 0.2 - 0.65 | | 2.94 | Singh *et al.* 1995 |
| | Pacific Ocean (10-45°N, 100-230°E) | 0 -2 | 0.47 ± 0.01 | | -1.78 x 10$^{-4}$ | winter/ spring 2011 | Singh *et al.* 2003 |
| | Pacific Ocean (10-45°N, 100-230°E) | 2- 4 | 0.64 ±0.21 | | -1.78 x 10$^{-4}$ | winter/ spring 2011 | Singh *et al.* 2003 |
| | North Atlantic, Mace Head | 0.025 | 0.5 | 0.16 - 1.67 | | 7.-9.2002 | Lewis *et al.* 2005 |
| | Tropical Atlantic | 0.018 | 0.53 | | 8.5 | 10.-11. 2002 | William *et al.* 2004 |
| | Wes. Trop. Atlantic O. | 0.01 | 0.36 ±0.051 | | - 0.1 – -15[b] | 5.-7.2004 | Marandino *et al.* 2005 |
| | Pacific O. (10-50 °N) | 0.1-12 | 0.44 ±0.2 | | | 2.-4.2001 | Singh *et al.* 2004 |
| | South China Sea (14 - 25°N, 113-125°E) | | 0.45 ±0.18 | | | all year 2006 | Elias *et al.* 2011 |
| | South China Sea (14 - 25°N, 113-125°E) | | 0.45 ±0.24 | | | 2007 | Elias *et al.* 2011 |
| | Pacific (35°N-35°S, 90°W-150°E) | up to 12 | | 0.35 -0.6 | | 3.-4.1999 | Sing *et al.* 2001 |
| | Atlantic O. (50°N - 50°S, 10-60°W) | 0.018 | | > 0 - 0.9 | - 0.2 ±2.5[b] | 10.-11.2012 | Yang *et al.* 2014 (a) |
| | Atlantic (65°N - 40-70°W) | | | 0.1 - 1.1 | - 11 ±5[b] | 10.-11. 2013 | Yang et al. 2014 (b) |
| | Caribbean Sea | 0.01 | 0.4 ±0.15 | 0.2 - 1 | | 10.1988 | Zhou and Mopper, 1993 |
| | 100 km East of Bahamas | 0.01 | | | 23.3 | 3.1989 | Zhou and Mopper, 1997 |
| | North Atlantic, Mace Head | 0.025 | | 0.24 – 4.9 | | 10.1993 – 4.1995 | Solberg *et al.* 1996 |
| | Indian Ocean | 0.028 | 0.6 ±63 | 0.45 – 2.4 | | 3.1999 | Wisthaler *et al.* 2002 |
| Propa-nal | Pacific Ocean (10-45°N, 100-230°E) | 0 -2 | 0.07 ±0.02 | | 6 x 10$^{-4}$ | winter/ spring 2011 | Singh *et al.* 2003 |
| | Pacific Ocean (10-45°N, 100-230°E) | 2- 4 | 0.06 ±0.02 | | | winter/ spring 2011 | Singh *et al.* 2003 |
| | Pacific O. (10-50 °N) | 0.1-12 | 0.04 ±0.02 | | | 2.-4.2001 | Singh *et al.* 2004 |





| Buta-none | Pacific O. (10-50 °N) | 0.1-12 | 0.03 ±0.03 | | 2.-4.2001 | Singh *et al.* 2004 |
|---|---|---|---|---|---|---|
| | Trop. Wes. Atl. O. (10- 30°N, 60 - 80°W) | 0.01 | 0.03 | | 10.1998 | Zhou and Mopper, 1993 |

[a] negative values refer to a flux from the atmosphere into the ocean, positive values indicate fluxes out of the ocean
[b] Direct flux was measured using eddy covariance

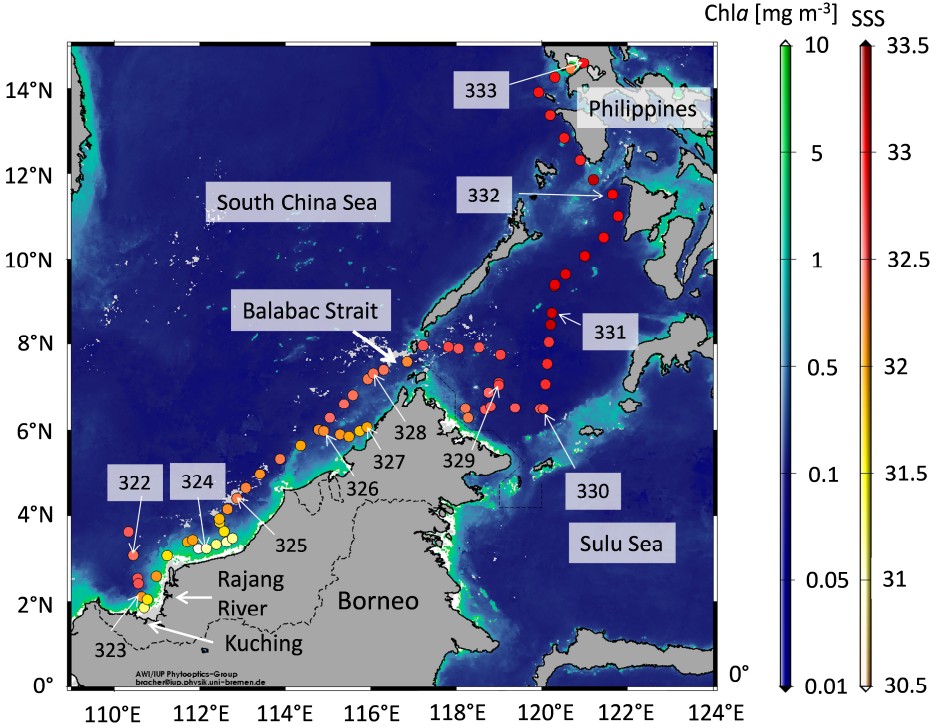

Figure 1: Cruise track (17-29 November 2011) showing salinity in the surface seawater at each point when an underway measurement was conducted. Numbers show the day of the year and the arrows the location when a day started. Background colors show chlorophyll a (Chl*a*) (level-2) data from MERIS satellite sensor (onboard Envisat) using the Polymer-product (Steinmetz et al., 2011). Note that the Chl*a* concentrations are shown on a logarithmic scale.





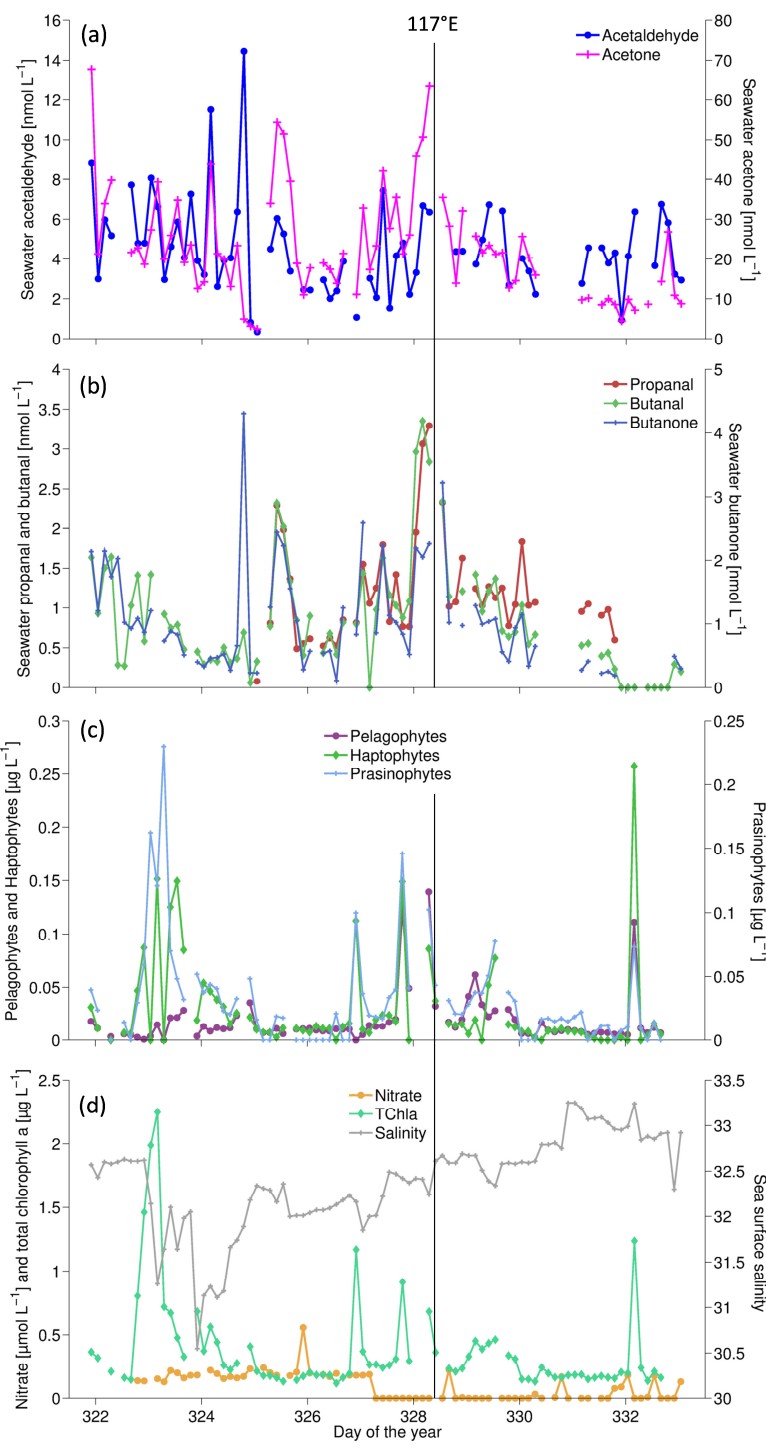

Figure 2: OVOC concentrations in water, the phytoplankton groups pelagophytes, haptophytes (mainly *Phaeocystis spp.*) and prasinophytes, nitrate, chlorophyll a and salinity plotted against the day of the year



(DOY). Location of the data points can be deduced from Fig. 1. The vertical line shows the location of the Balabac Strait (117° E), where coastal conditions changed to open ocean conditions.

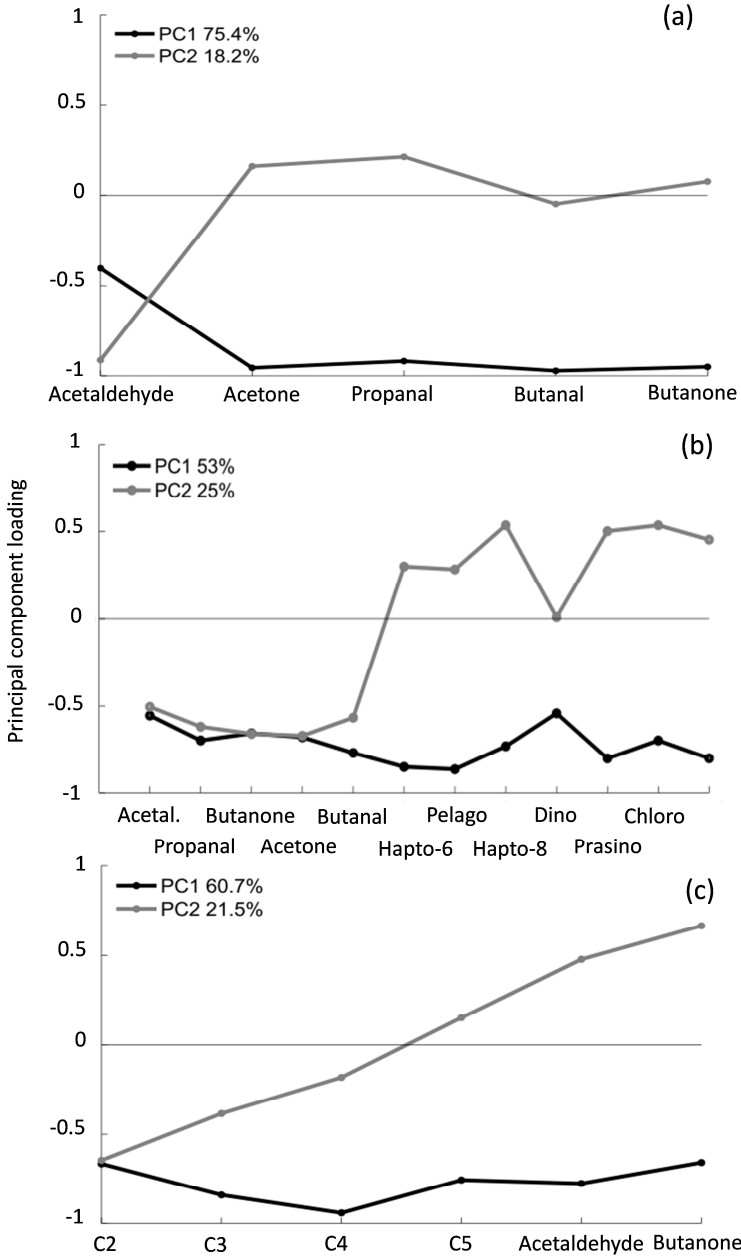

5      Figure 3: Principal component analysis (PCA) of OVOCs in seawater (panel a); between OVOCs in seawater and phytoplankton (panel b); and between fluorescent dissolved organic matter (FDOM) groups, acetaldehyde, and butanone in seawater (panel c). Numbers on the y-axes indicate the factor loadings of each variable of each




principal component (PC). The percentages show the explained variability of the dataset by each PC. Hapto-6 = haptophytes, mainly coccolithophorids; Pelago = pelagophytes; Hapto-8 = haptophytes, mainly *Phaeocystis spp.*; Dino = dinoflagellates; Prasino = prasinophytes; Chloro = chlorophytes.

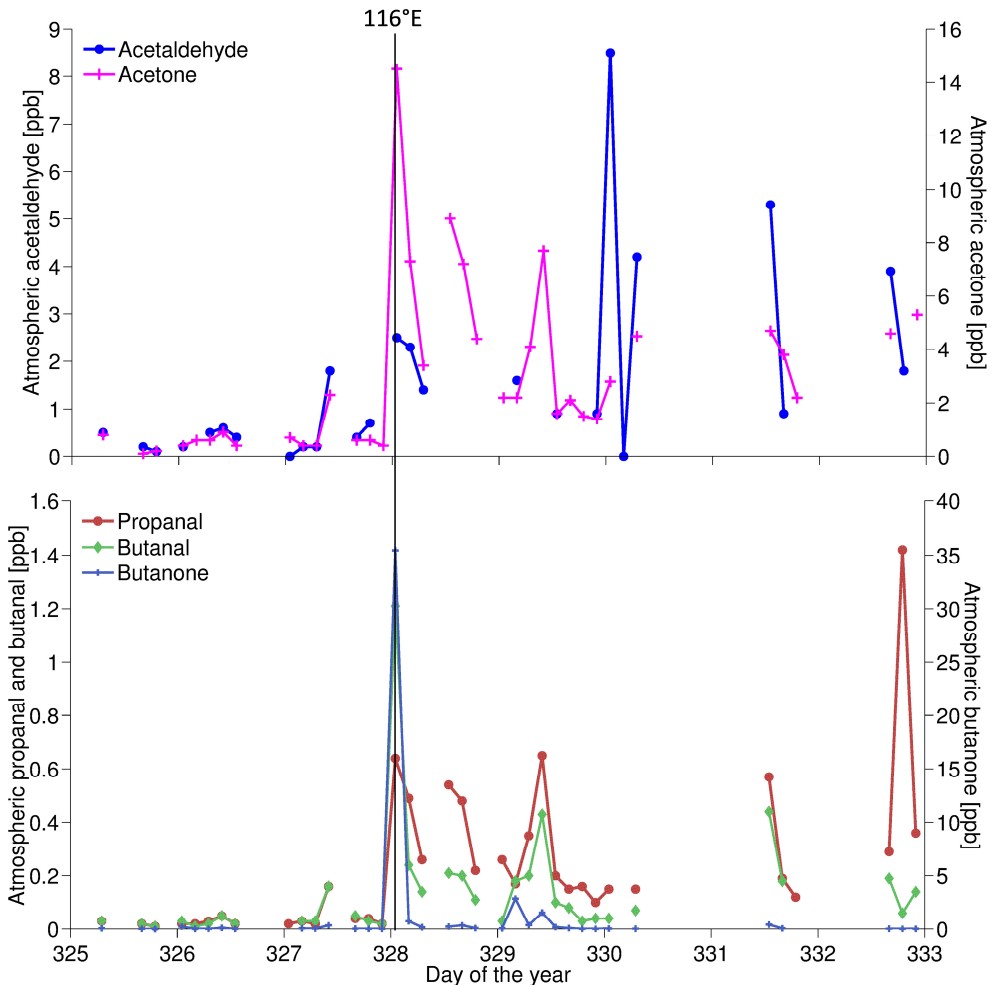

Figure 4: OVOC concentrations in the air plotted against the day of the year (DOY). Please note that the measurements of the atmospheric data started three days later (on day 325) than the water measurements. Vertical line indicates the location of 116° E.



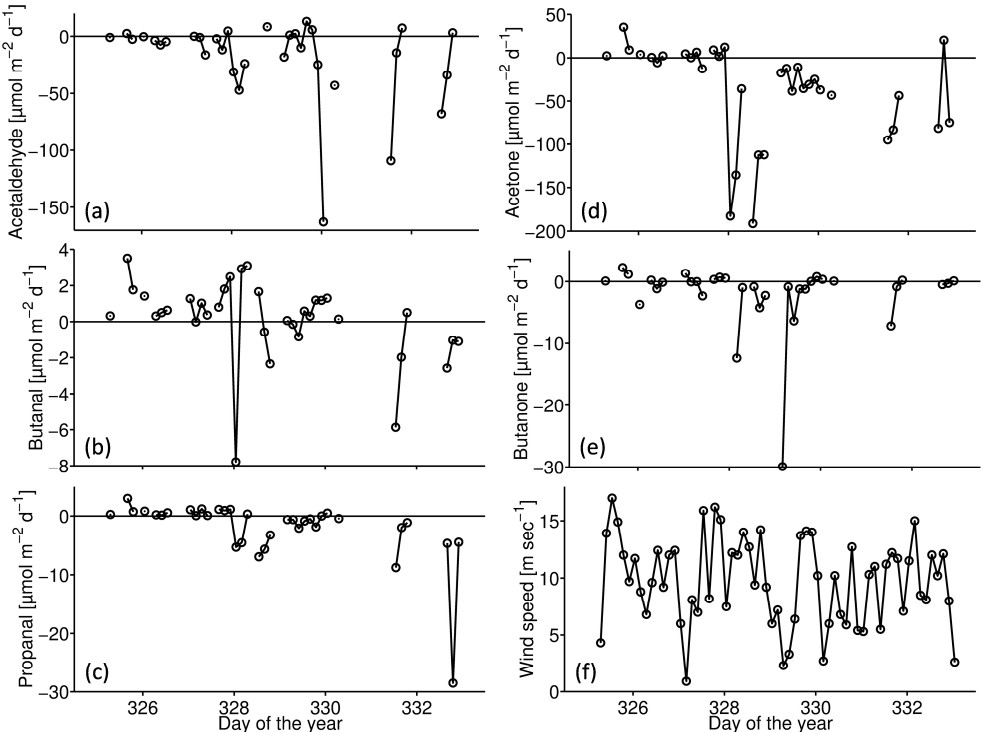

Figure 5: Calculated fluxes of the OVOCs (panel a-e). Fluxes above the zero line indicate fluxes out of the ocean, fluxes below the zero line indicate fluxes into the ocean. Panel f shows the wind speed plotted against the day of the year (DOY).

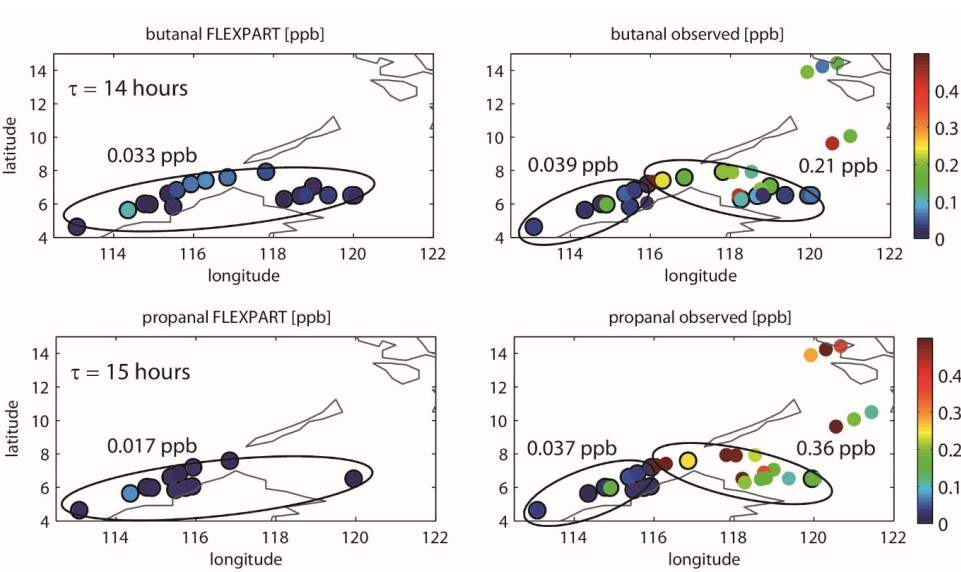





Figure 6: Modelled (left panels) and observed (right panels) atmospheric mixing ratios of butanal (upper panels) and propanal (lower panels) at the location of the respective measurement. Model simulations have been carried out for all locations where the flux was from the ocean into the atmosphere (also marked by a black edge around the symbols). Modelled mixing ratios were averaged over all data points (left panels) indicated by the number in ppb. Observed mixing ratios were averaged separately east and west of 116° E (right panels, two black circles). Atmospheric lifetimes (τ) are given in the left panels.

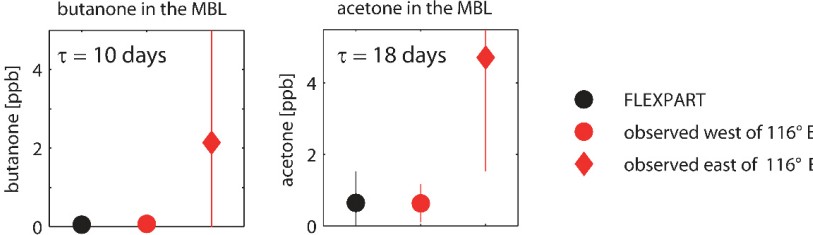

Figure 7: Mean modelled (averaged over all data points, black dot) and observed (averaged east (red dot) and west (red diamond) of 116° E) atmospheric mixing ratios of butanone (left panel) and acetone (right panel) are shown. The atmospheric lifetimes (τ) are given in the panels.

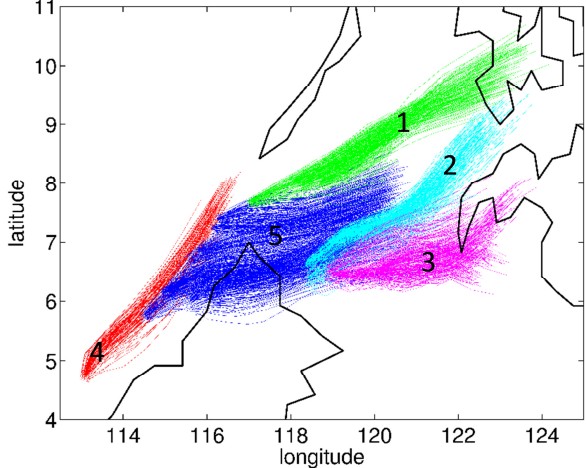

Figure 8: Backward trajectories over 24 hours from all OVOC measurement locations are shown. The trajectories are color-coded according to the atmospheric transport regime.





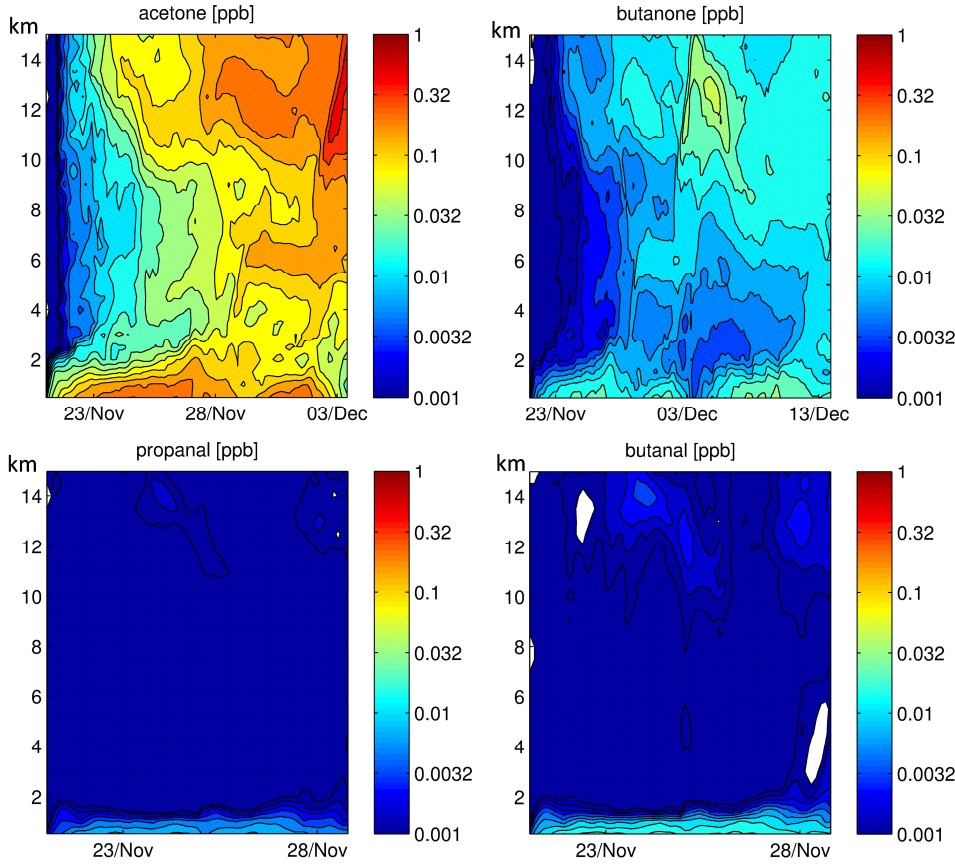

Figure 9: Vertical distribution of the OVOCs derived from FLEXPART model simulations. OVOCs are released from the measurement locations over the time period of the cruise according to the respective ocean-atmosphere flux.

