# Peer review of "Oxygenated volatile organic carbon in the western Pacific convective centre: ocean cycling, air-sea gas exchange and atmospheric transport"

_Atmospheric Chemistry and Physics, 2017_

## Referee Comment (RC1) · M. Yang (Referee) · 30 Mar 2017

This is a really interesting paper that describes a set of oxygenated volatile organic compound (OVOC) measurements in surface ocean/lower atmosphere of the western Pacific. The authors explored relationships between seawater OVOC concentrations and phytoplankton groups as well as types of dissolved organic matter. They also constrained 1) the contribution of air-to-sea flux on the oceanic OVOC budget, 2) the contribution of sea-to-air flux on the atmospheric OVOC budget, and finally 3) the contribution of boundary layer OVOCs to the upper troposphere (relevant for HOx cycling). The number of data points is somewhat limiting (especially the atmospheric OVOC concentrations, which are necessary for the flux calculation), but the analysis is sound

and the data interpretation seems reasonable. I think the paper is publishable after minor revisions.

Specific comments:

p.2 Line 6-31. To be more precise, the authors should specify the sources and sinks of individual OVOC compounds in these 2 paragraphs, rather than always using the term "OVOCs".

p.3 line 4-10. The 'either...or' scenario doesn't completely hold due to photochemical destructions of OVOCs, for example. This paragraph also feels disconnected from the neighboring paragraphs. Can probably just move it to Section 3.2.2.

p.4 line 5-19. Did the authors thoroughly test the effect of using potassium carbonate as the drying agent – that it doesn't cause any loss/contamination in OVOCs?

Out of curiosity, in this setup how much of each OVOC compounds is purged out from the water phase after 20 minutes approximately?

The fact that the water standards were prepared in Milli Q water while the actual samples were seawater will likely create a small bias in the measurement (due to the effect of salinity on gas solubility, and thus purging efficiency)

Given the analytical errors and reproducibility stated, what are the limits of detections for these compounds in seawater?

Line 21-31. Was potassium carbonate used to dry the air samples too?

p. 6, line 11-19. As discussed by Yang et al PNAS 2013 (www.pnas.org/content/110/50/20034.abstract), the Duce et al. 1991 parameterization probably underestimates airside diffusional resistance and thus overestimates total airside transfer velocity.

p. 10 line 24. If atmospheric OVOC input is important for the ocean mixed layer OVOC budget, some of the spatial variability/correlations in seawater OVOC can preassembly be in part explained by variability/correlations in atmospheric deposition fluxes of OVOCs?

Have the authors looked for correlations of OVOCs in air? By eye some of the compounds appear to strongly correlate.

p. 10 line 39-40. This sentence contradicts with what's stated in your abstract, that most of the OVOC fluxes were into the ocean. I'd pick a tone and stay with it.

Also, do the OVOC seawater concentrations, atmospheric concentrations, and air-sea fluxes vary diurnally? If so, it might provide further hints to the possible sources.

p. 11 line 23 (typo here, should be Section 2.8, not 2.6)

p. 12, line 31-33. This is reasonable, though it's still possible that some unmeasured oceanic hot spots upwind of the eastern region were contributing to the atmospheric OVOCs.

p. 12, line 40. Are only positive sea-to-air fluxes used here (as implied by the omission of acetaldehyde), or do the average air-sea fluxes include both positive and negative fluxes? In the case of the former, this calculation would represent the maximum contribution to UT.

More generally, I think the authors can benefit from differentiating between net vs. gross fluxes here (i.e upwards and downwards; see Yang et al. ACP 2014). Even in a case where the OVOCs are in air-sea equilibrium, molecules of OVOCs from the sea will still emit into the atmosphere and molecules of OVOCs in the atmosphere will still dissolve into the sea (just the two fluxes cancel out mathematically). It could be that the molecules of OVOCs in air and in water come from rather different sources. Long story short, to look at the upper-limit impact of ocean emissions on the UT, I think it'd be useful to do another FLEXPART model run prescribed with the gross emission fluxes (= K * Cw). Of course, for the ketones that have reasonably long lifetimes, some molecules of ketones in the atmosphere will have come from the ocean and vice versa.

Figure 9. to make it easier for the readers, can the authors please stick to one timestamp format (either day of year or day/month)?

[Figure]

---

## Referee Comment (RC2) · Anonymous Referee #2 · 29 Jun 2017

General comments:

The authors reported shipborne observations of a series of oxygenated VOCs (OVOCs) in the western tropical Pacific. The spatial and temporal distributions and the air-sea fluxes of five OVOCs were presented and discussed. For some species, these data were published for the first time. I appreciate their effort. The authors also tried to explore the uplifting of these OVOCs to the upper troposphere by using the FLEXPART model. I found the paper well organized and written in general, putting both atmospheric and oceanic perspectives. This paper would be a nice piece of work contributing to the groups looking at the air-sea exchange of organics. In addition to

the comments from another reviewers (with which I agree on most comments), I only have several minor and/or technical comments that the authors can consider before publication, as listed below.

Minor comments:

FLEXPART analysis, Fig 6-9: To be honest I found the analysis with FLEXPART is a bit premature and resulting implications are speculative, as the data number is so limited and the authors cannot deal with mixing with the Lagrangian model, while the authors noted it (P11, L26-28). My question here is how well the Lagrangian-type model works in this hot and humid atmosphere in the tropics. Backward trajectories often fail in tropical MBL, so I wonder if there is the same issue or not.

P1, L32: "relatively" high

P8, L36: Did Whelan et al. test both macroalgae and phytoplankton, and find that only macroalgae produced VOCs? Or did they only test macroalgae? Please clarify.

P11, L1: The "on" average

P11, L23: "release trajectories" sounds a bit odd to me, perhaps say "release particles" or "start trajectories"? Anyway consider to rephrase.

P11, L6-8; Table 3: One literature data is missing here. There is a literature data of acetone flux in the western North Pacific by the gradient method. Tanimoto et al. (2014) reported the acetone flux to be $2.7 \pm 1.3$ $\mu$mol/m2/day, for the western North Pacific (15-20N, 137E) in 2010. Please add it into here.

Reference:

Tanimoto, H., S. Kameyama, T. Iwata, S. Inomata, Y. Omori, Measurement of air-sea exchange of dimethyl sulfide and acetone by PTR-MS coupled with gradient flux technique, Environ. Sci. Technol., 48, 526-533, 2014.

---

## Author Comment (AC1) · 28 Jul 2017

Response to the referee#1 Mingxi Yang

We thank Mingxi Yang (MY) for the intensive work on our manuscript and the helpful comments. We appreciate his effort to improve our manuscript.

MY: This is a really interesting paper that describes a set of oxygenated volatile organic compound (OVOC) measurements in surface ocean/lower atmosphere of the western Pacific. The authors explored relationships between seawater OVOC concentrations and phytoplankton groups as well as types of dissolved organic matter. They also

constrained 1) the contribution of air-to-sea flux on the oceanic OVOC budget, 2) the contribution of sea-to-air flux on the atmospheric OVOC budget, and finally 3) the contribution of boundary layer OVOCs to the upper troposphere (relevant for HOx cycling).

The number of data points is somewhat limiting (especially the atmospheric OVOC concentrations, which are necessary for the flux calculation), but the analysis is sound and the data interpretation seems reasonable. I think the paper is publishable after minor revisions.

Specific comments: p.2 Line 6-31. To be more precise, the authors should specify the sources and sinks of individual OVOC compounds in these 2 paragraphs, rather than always using the term "OVOCs".

Authors: We have changed the text to give some specific examples of different OVOCs for the different sources and sinks. For instance: "OVOCs, such as acetone and acetaldehyde, are involved in the production of reactive nitrogen compounds, such as nitrogen dioxide (NO2, involved in ozone production), peroxynitric acid (HNO4), and nitric acid (HNO3), and they are precursors of peroxyacetyl nitrate (PAN), a persistent harmful pollutant (Folkins and Chatfield, 2000;Fischer et al., 2014). OVOCs, such as acetone, are a source for hydrogen oxide radicals (HOx), which is of special importance for the upper troposphere (UT), where the concentration of a main precursor, namely water vapor, is much lower than at the Earth surface (Singh et al., 1995;Wennberg et al., 1998;Müller and Brasseur, 1999). Furthermore, OVOCs, such as acetone, acetaldehyde and propanal, can contribute to particle formation in the atmosphere, resulting in albedo enhancement (Blando and Turpin, 2000). ..." However, most of the OVOCs have the same or similar sources; thus, we kept often the term OVOCs to keep the sentences simple. We hoped that providing the appropriate references would aid in this. Furthermore, acetone and methanol are the most well studied OVOCs and the literature refers mainly to these two compounds. There were less specific publications about e.g. butanone or butanal compare to acetone and methanol. MY: p.3 line 4-10. The 'either...or' scenario doesn't completely hold due to photochemical destructions

of OVOCs, for example. This paragraph also feels disconnected from the neighboring paragraphs. Can probably just move it to Section 3.2.2. Authors: Indeed, the OVOCs are not only distributed in the atmosphere as indicated by the "either…or" scenario, but can also be destroyed and produced. Thus, we rewrote the first sentence (and subsequent sentences). However, we think it should stay in the introduction, in order to give some background on the region as a transport pathway to the UT. We added another sentence linking this paragraph to the beginning of the Intro, to underline the importance of OVOCs as an UT HOx source. .

MY: p.4 line 5-19. Did the authors thoroughly test the effect of using potassium carbonate as the drying agent – that it doesn't cause any loss/contamination in OVOCs?

Authors: We tested a couple of different methods to dry the sampling gas stream to avoid blocking our cryotrap. We tested different cold traps, such as ethanol mixed with dry ice or frozen water mixed with different salts. However, these cold traps were not cold enough to freeze the water fast enough without losing the OVOCs. The OVOCs dissolved immediately in the water when it condensed in the water trap. We also tried Nafion. However, ketones like acetone were trapped in Nafion. Only potassium carbonate showed good reproducibility without losing significant amounts of OVOCs. We had to pretreat the K2CO3 before use by flushing it with helium for 20min. When the K2CO3 got too wet OVOCs were trapped again, so we had to replace the trap after around 5 measurements. We now give more details about this trap in the text.

MY: Out of curiosity, in this setup how much of each OVOC compounds is purged out from the water phase after 20 minutes approximately?

Authors: We never did a full sparge of the samples to see how long it took to purge approximately 100%. It took too long and flooded the lines with too much water. We did repeated tests for reproducibility of the signal with the same standard concentration at different sparging times. Therefore, we have confidence in the measurements, but cannot answer what percent was sparged i in 20 mins.

MY: The fact that the water standards were prepared in Milli Q water while the actual samples were seawater will likely create a small bias in the measurement (due to the effect of salinity on gas solubility, and thus purging efficiency).

Authors: We discussed this at length during the time of setting up our system. We tested adding our standard solution to seawater (SW) vs. MilliQ (MQ) several times. The seawater standards were never very repeatable/precise. For example, after sparging both SW and MQ for approx. 3 days, the blanks in seawater were at least double the MQ blanks. Eventually, the uncertainty in our calibrations was larger than the possible systematic problem of the salting out effect. We could control our MQ blanks better and had much better reproducibility, so we decided to use MQ and take the highest uncertainties.

MY: Given the analytical errors and reproducibility stated, what are the limits of detections for these compounds in seawater?

Authors: The detection limit of all OVOCs was around 0.06 nmol L-1. We added this info in the text.

MY: Line 21-31. Was potassium carbonate used to dry the air samples too?

Authors: Yes. Also here we tested the possible loss and contamination of OVOCs. We used the same setup for air samples as for water samples to be consistent.

MY: p. 6, line 11-19. As discussed by Yang et al PNAS 2013 (www.pnas.org/content/110/50 /20034.abstract), the Duce et al. 1991 parameterization probably underestimates airside diffusional resistance and thus overestimates total airside transfer velocity.

Authors: We tested our fluxes by comparing them to a newly computed flux using Johnson et al. (2010)'s recommendation for ka (specifically Eqs. 10, 15, 16). The values are, on average, 20% higher (lower if fluxes are negative) using the new method. Since the values we report in the original manuscript are lower, they are a more conservative estimate of the flux. We will keep these values but make a statement in the text reflecting this point and adding 20% to our uncertainty estimate.

MY: p. 10 line 24. If atmospheric OVOC input is important for the ocean mixed layer OVOC budget, some of the spatial variability/correlations in seawater OVOC can preassembly be in part explained by variability/correlations in atmospheric deposition fluxes of OVOCs?

Authors: We correlated also seawater OVOCs with OVOCs in the atmosphere and couldn't find any significant correlations. We assume that the variability in seawater and atmosphere due to the complex and different controlling factors in both spheres is too high to find similarities of marine and atmospheric OVOCs.

MY: Have the authors looked for correlations of OVOCs in air? By eye some of the compounds appear to strongly correlate.

Authors: That's a good point. We correlated the atmospheric OVOCs now and found significant correlation between acetone and propanal (r2 = 0.84) and between acetone and butanal (r2 = 0.72). We added this to the text in Section 3.2.1. We have been careful in the manuscript to not read too much into the fact that not all gases are correlated among each other, since atmospheric measurements have been taken less often (compared to oceanic measurements) and thus detecting significant correlation is more difficult. Therefore, one cannot exclude that the other gases are also controlled by the same factors, even though they do not show significant correlations

MY: p. 10 line 39-40. This sentence contradicts with what's stated in your abstract, that most of the OVOC fluxes were into the ocean. I'd pick a tone and stay with it.

Authors: MY is right that it sounds contradictory. We changed this sentence and the one in the abstract to make clear that we calculated, on average, a negative flux. Just to be clear, the fluxes were quite small, showing that the ocean and the atmosphere were near equilibrium for OVOCs.

[Figure]

MY: Also, do the OVOC seawater concentrations, atmospheric concentrations, and air-sea fluxes vary diurnally? If so, it might provide further hints to the possible sources.

Authors: We compared the data we sampled in the night with the data from the day. However, we couldn't find significant differences between night and day datasets, probably do to different regions the day and night samples were taken. Unfortunately, we did not have 24h drift stations during the cruise to observe diurnal cycles of OVOCs at one place. It is worth to investigate this on the next cruise with drift stations.

MY: p. 11 line 23 (typo here, should be Section 2.8, not 2.6)

Authors: Thanks, we changed it.

MY: p. 12, line 31-33. This is reasonable, though it's still possible that some un-measured oceanic hot spots upwind of the eastern region were contributing to the atmospheric OVOCs.

Author: MY is right that marine hot spots might have occurred at some distance to the cruise track and that we could not measure. , Given the high variability of the measured OVOCs in water, the existence of further hot spots is of course possible. We made a short statement to include this possibility. However, we think this is less likely, as these hotspots would need to be much larger than anything we did measure during the cruise in order to explain atmospheric mixing ratios 10 to 20 times larger than the ones resulting from the observed oceanic sources..

MY: p. 12, line 40. Are only positive sea-to-air fluxes used here (as implied by the omission of acetaldehyde), or do the average air-sea fluxes include both positive and negative fluxes? In the case of the former, this calculation would represent the maximum contribution to UT.

Authors: The fluxes we report are net fluxes, as they are computed using the net concentration gradient.

MY: More generally, I think the authors can benefit from differentiating between net

vs. gross fluxes here (i.e upwards and downwards; see Yang et al. ACP 2014). Even in a case where the OVOCs are in air-sea equilibrium, molecules of OVOCs from the sea will still emit into the atmosphere and molecules of OVOCs in the atmosphere will still dissolve into the sea (just the two fluxes cancel out mathematically). It could be that the molecules of OVOCs in air and in water come from rather different sources. Long story short, to look at the upper-limit impact of ocean emissions on the UT, I think it'd be useful to do another FLEXPART model run prescribed with the gross emission fluxes (= K * Cw). Of course, for the ketones that have reasonably long lifetimes, some molecules of ketones in the atmosphere will have come from the ocean and vice versa.

Authors: We thank MY for this point and discussed it heavily, but finally came to the conclusion that using gross fluxes out of the ocean is not very realistic for determining atmospheric budgets. The only way that the gross flux could impact the UT OVOC budget is if the vertical transport out of the marine boundary layer would take place on shorter time scales than the two-way ocean-atmosphere exchange. However, this is unrealistic and such an assumption would bias our assessment of the amount of OVOCs that make it to the UT Therefore, after giving this point considerable thought, we decided that the net flux is a better choice for determining a realistic impact of OVOCs on the UT.

MY: Figure 9. to make it easier for the readers, can the authors please stick to one timestamp format (either day of year or day/month)?

Authors: Thanks for the suggestion, we have changed the Figure.

---

## Author Comment (AC2) · 28 Jul 2017

Response to the anonymous referee #2

We thank the referee #2 for the effort to help improve our paper.

Referee #2: General comments: The authors reported shipborne observations of a series of oxygenated VOCs (OVOCs) in the western tropical Pacific. The spatial and temporal distributions and the air-sea fluxes of five OVOCs were presented and discussed. For some species, these data were published for the first time. I appreciate their effort. The authors also tried to explore the uplifting of these OVOCs to the upper

troposphere by using the FLEXPART model. I found the paper well organized and written in general, putting both atmospheric and oceanic perspectives. This paper would be a nice piece of work contributing to the groups looking at the air-sea exchange of organics. In addition to the comments from another reviewers (with which I agree on most comments), I only have several minor and/or technical comments that the authors can consider before publication, as listed below.

Minor comments:

Referee #2: FLEXPART analysis, Fig 6-9: To be honest I found the analysis with FLEX-PART is a bit premature and resulting implications are speculative, as the data number is so limited and the authors cannot deal with mixing with the Lagrangian model, while the authors noted it (P11, L26-28). My question here is how well the Lagrangian-type model works in this hot and humid atmosphere in the tropics. Backward trajectories often fail in tropical MBL, so I wonder if there is the same issue or not.

Authors: The Lagrangian-type models work well for the meteorological conditions found during the SHIVA campaign. A perfluorocarbon tracer system, specifically designed for Lagrangian aircraft experiments, has been successfully applied during the SHIVA campaign (Ren et al., 2015). The atmospheric distribution of the tracer, released from the RV Sonne, was simulated with the Lagrangian model HYSPLIT and probed by the research aircraft Falcon giving a good agreement with some differences in the plume dispersion. FLEXPART backward trajectories are in good agreement with the HYS-PLIT trajectories from Ren et al. (2015) suggesting an overall realistic simulation of the computed air mass transport over the 24 hour time period. This information has been added to the manuscript. It is true that the Lagrangian-type models cannot deal with mixing between air parcels throughout the boundary layer. On the other hand, we have only very limited OVOC source data and, therefore, have to exclude mixing in our approach. However, we do not agree that the results are speculative. This study does not aim to estimate the exact horizontal OVOC distribution, but only the maximum atmospheric mixing ratios that could result from the observed oceanic sources. To answer

this question, the indirect assumption of constant fluxes over time and space (equivalent to no mixing) can be made. The question of how well the simulated trajectories described the air mass transport is only important for the analysis of the backward trajectories, which agree well with the above mentioned tracer experiment,

Ren, Y., Baumann, R., and Schlager, H.: An airborne perfluorocarbon tracer system and its first application for a Lagrangian experiment, Atmos. Meas. Tech., 8, 69-80, https://doi.org/10.5194/amt-8-69-2015, 2015.

Referee #2: P1, L32: "relatively" high

Authors: We changed it.

Referee #2: P8, L36: Did Whelan et al. test both macroalgae and phytoplankton, and find that only macroalgae produced VOCs? Or did they only test macroalgae? Please clarify.

Authors: They tested both macroalgae (4 species in total, Ulva lactuca, Ascophyllum nodosum, Gracilaria tikvahiae, Hypnea musciformis) and phytoplankton (4 species in total, Thalassiosira sp., Gymnodinium sp., Emilinia huxleyi, Skelotenema costatum) using the same technique for all cultures. They found only in macroalgae cultures significant production of VOCs such as acetone, propanal, butanal and 2-butanone. We clarified this in the text.

Referee #2: P11, L1: The "on" average

Authors: We rewrote this part of the paper to clarify the apparent contradiction between the abstract and this paragraph, as requested by the other referee. We wrote instead: "The fluxes into the ocean are caused by localized, strong sinks such as observed in the Balabac Strait . . ."

Referee #2: P11, L23: "release trajectories" sounds a bit odd to me, perhaps say "release particles" or "start trajectories"? Anyway consider to rephrase.

Authors: We have changed the text to 'start trajectories', thanks for the suggestion.

Referee #2: P11, L6-8; Table 3: One literature data is missing here. There is a literature data of acetone flux in the western North Pacific by the gradient method. Tanimoto et al. (2014) reported the acetone flux to be 2.7 _ 1.3 _mol/m2/day, for the western North Pacific (15-20N, 137E) in 2010. Please add it into here. Reference: Tanimoto, H., S. Kameyama, T. Iwata, S. Inomata, Y. Omori, Measurement of air-sea exchange of dimethyl sulfide and acetone by PTR-MS coupled with gradient flux technique, Environ. Sci. Technol., 48, 526-533, 2014.

Authors: We included this reference in table 3. Thanks for advising us of this reference.

———————————————————

---

## Editor Decision (ED1)

**Editors comment paper acp-2017-9; Oxygenated volatile organic carbon in the western Pacific convective centre: ocean cycling, air-sea gas exchange and atmospheric transport by Schlundt et al.**

Abstract: "The flux of atmospheric OVOCs was on average into the ocean for all gases, except butanal, with a few important exceptions near the coast of Borneo"

This added sentence was actually raising confusion since 1) the previous sentences are initially suggesting that the ocean is a source of OVOCs 2) are the few exceptions referring to butanal or to the fact that for all OVOCS there is generally deposition? I would anyhow also reformulate this sentence after having read in detail again the overall the document. It seems that those couple of sentences of the abstract were not clearly explaining the main findings. I propose just a change in the sequence of the sentences that might help in overcoming this:

"The measurement-inferred OVOCs fluxes away from the North Borneo coastal waters were generally reflecting uptake of OVOCs by the ocean for all gases, except of butanal. Over the Borneo coastal waters, the atmospheric OVOC mixing ratios were relatively high compared with literature values, suggesting that this coastal region of North Borneo is a local hotspot for atmospheric OVOCs including a significant coastal water source of atmospheric OVOC's."

Introduction:

Reading the following sentence;

"OVOCs, such as acetone and acetaldehyde, are involved in the production of reactive nitrogen compounds, such as nitrogen dioxide ($NO_2$, involved in ozone production), peroxynitric acid ($HNO_4$), and nitric acid ($HNO_3$)"

I started to wonder if you can really say that $NO_2$ is produced involving these OVOCs. The $NO_2$ is produced from the NO involving the $RO_2$ and which is affected by the OVOCs but the $NO$-$NO_2$-$O_3$ system is expressing a cycle. I would rephrase this to:

"OVOCs, such as acetone and acetaldehyde, are affecting the cycling of the reactive nitrogen compounds nitrogen oxide (NO) and nitrogen dioxide ($NO_2$), and associated ozone production, and involved in the production of peroxynitric acid ($HNO_4$), and nitric acid ($HNO_3$)"

Pp2: Carpenter *et al.* (2012) and references therein

Make this reference listing consistent with how other references are included.

Pp2; line 37: -48 to -1 Tg $yr^{-1}$; does a negative value here reflect a source or sink for OVOCs to the atmosphere? I would add after the listed references ", with the negative values here reflecting the ocean being a sink for acetone"

Pp3, line 2 "….no ocean-atmosphere butanal or butanone fluxes.."

Pp3, line 8 "..trace gases  into the UT…"

Pp 6, line 19; add here something like "Note that according to Eqn 1, a negative flux reflects

a flux from the atmosphere to the ocean and vice versa."

Pp 6, line 23: "…at 10 m height and on.."
and replace "Within the Johnson (2010) publication, there is a critical discussion of using Duce et al. (1991) to compute $k_a$" with "Johnson (2010) provided a discussion of using Duce et al. (1991) to compute $k_a$"

Pp 6, line 25: "The newly computed fluxes were  on average  20% higher (lower in the case of negative fluxes). We treat this difference in the calculated fluxes as uncertainty and use the lower fluxes as a conservative estimate of OVOC fluxes into and out of the ocean surface." This revised text further confuses the interpretation of the paper in terms what negative and positive values of fluxes reflect. So, if I get it right using the alternative approach to calculate $k_a$, inferred negative/deposition (atmosphere-to ocean) fluxes are reduced by about 20% whereas positive fluxes (emissions) are "on average 20% higher"?

Pp6; line 35 and beyond; since you are discussing the emissions of OVOCS into the MBL being studied with FLEXPART; you list all the processes that are considered in FLEXPART except of emissions! How are these treated in this model? As a negative dry deposition flux?

Pp 7, line 30: "compared to a study along the South-East Florida coast" (or alternatively "in the coastal waters of South-East Florida"

Pp 8; line 16 "compared with"; check the whole document actually on this; according to me it is here "compared to" (compared with is used when things are similar, e.g., of magnitude) whereas "compared to" is used when you contrast things.

Pp 10: line 28 "for the entire ocean mixed layer" (to contrast this with the atmospheric mixed/boundary layer)

Pp 11, line 30: suggesting to change to; "For all measurement locations with a positive flux, reflecting the ocean being a source for atmospheric OVOCs," and would it be useful here to shortly indicate how many of all samples are indeed showing positive fluxes?

Pp 12: line 30: "…driven by the hotspots east of 116°E which occur in all three OVOCs.." this statement reads weird: I would suggest to say, "..due to presence of areas east of 116 °E with large sources reflected in the measurements of all three OVOCs"

Pp 13: line 9-10: " hot spots exist at some short distance from the cruise track area. However, we think this is less likely as these hotspots"; hot spots or hotspots?
And would it not be better to refer instead of a hotspot to " a strong source area"?

Pp 13, line 35: list the reference Bliztz et al. in the proper way

Pp 13, line 41: I happen to know the EMAC modelling system but not other readers; revise by or explaining the acronym or simply referring to EMAC as a "global chemistry-climate modelling system"

---

## Author Response (AR2)

Authors: Thanks a lot to the Editor to read our paper again and helped us to improve our manuscript.

Editors comment paper acp-2017-9; Oxygenated volatile organic carbon in the western Pacific convective centre: ocean cycling, air-sea gas exchange and atmospheric transport by Schlundt et al.

Abstract: "The flux of atmospheric OVOCs was on average into the ocean for all gases, except butanal, with a few important exceptions near the coast of Borneo"

This added sentence was actually raising confusion since 1) the previous sentences are initially suggesting that the ocean is a source of OVOCs 2) are the few exceptions referring to butanal or to the fact that for all OVOCS there is generally deposition? I would anyhow also reformulate this sentence after having read in detail again the overall the document. It seems that those couple of sentences of the abstract were not clearly explaining the main findings. I propose just a change in the sequence of the sentences that might help in overcoming this:

"The measurement-inferred OVOCs fluxes away from the North Borneo coastal waters were generally reflecting uptake of OVOCs by the ocean for all gases, except of butanal. Over the Borneo coastal waters, the atmospheric OVOC mixing ratios were relatively high compared with literature values, suggesting that this coastal region of North Borneo is a local hotspot for atmospheric OVOCs including a significant coastal water source of atmospheric OVOC's."

Authors: We changed the sentence as the Editor suggested with minor changes. We hope that the new version is clearer. We wrote: "The measurement-inferred OVOC fluxes showed generally an uptake of atmospheric OVOCs by the ocean for all gases, except for butanal. A few important exceptions were found along the Borneo coast, where OVOC fluxes from the ocean to the atmosphere were inferred. The atmospheric OVOC mixing ratios over the northern coast of Borneo were relatively high compared with literature values, suggesting that this coastal region is a local hotspot for atmospheric OVOCs."

Editor: Introduction: Reading the following sentence;

"OVOCs, such as acetone and acetaldehyde, are involved in the production of reactive nitrogen compounds, such as nitrogen dioxide (NO2, involved in ozone production), peroxynitric acid (HNO4), and nitric acid (HNO3)"

I started to wonder if you can really say that NO2 is produced involving these OVOCs. The NO2 is produced from the NO involving the RO2 and which is affected by the OVOCs but the NO-NO2-O3 system is expressing a cycle. I would rephrase this to:

"OVOCs, such as acetone and acetaldehyde, are affecting the cycling of the reactive nitrogen compounds nitrogen oxide (NO) and nitrogen dioxide (NO2), and associated ozone production, and involved in the production of peroxynitric acid (HNO4), and nitric acid (HNO3)"

Authors: The Editor is right that OVOCs only affecting the $NO_2$ cycle but not are involved in the production. We changed the sentence as the Editor suggested with slight changes for a better reading.

Editor: Pp2: Carpenter *et al.* (2012) and references therein. Make this reference listing consistent with how other references are included.

Autors: We changed it to "(Carpenter et al., 2012)".

Editor: Pp2; line 37: -48 to -1 Tg $yr^{-1}$; does a negative value here reflect a source or sink for OVOCs to the atmosphere? I would add after the listed references ", with the negative values here reflecting the ocean being a sink for acetone"

Authors: We added the part in the sentence.

Editor: Pp3, line 2 "....no ocean-atmosphere butanal or butanone fluxes.."

Pp3, line 8 "..trace gases  into the UT..."

Pp 6, line 19; add here something like "Note that according to Eqn 1, a negative flux reflects a flux from the atmosphere to the ocean and vice versa."

Authors: We changed it all.

Editor: Pp 6, line 23: "...at 10 m height and on.." and replace "Within the Johnson (2010) publication, there is a critical discussion of using Duce et al. (1991) to compute $k_a$" with

"Johnson (2010) provided a discussion of using Duce et al. (1991) to compute $k_a$"

Authors: done

Editor: Pp 6, line 25: "The newly computed fluxes were an on average of 20% higher (lower in the case of negative fluxes). We treat this difference in the calculated fluxes as uncertainty and use the lower fluxes as a conservative estimate of OVOC fluxes into and out of the ocean surface." This revised text further confuses the interpretation of the paper in terms what negative and positive values of fluxes reflect. So, if I get it right using the alternative approach to calculate $k_a$, inferred negative/deposition (atmosphere-to ocean) fluxes are reduced by about 20% whereas positive fluxes (emissions) are "on average 20% higher"?

Authors: We saw by using the ka from Duce 91 a 20% higher flux for positive fluxes and 20% lower flux for negative fluxes, which means that the fluxes were around 20% stronger in both directions. We changed it in the sentences as followed: "The newly computed fluxes were on average 20% higher for positive fluxes and around 20% lower in the case of negative fluxes, resulting in higher amount of OVOC concentrations exchanged between the ocean and the atmosphere in both directions. We treated this difference in the calculated fluxes as uncertainty and used the previous fluxes determined by using Duce 91 ka as a conservative estimate of OVOC fluxes into and out of the ocean surface."

Editor: Pp6; line 35 and beyond; since you are discussing the emissions of OVOCS into the MBL being studied with FLEXPART; you list all the processes that are considered in FLEXPART except of emissions! How are these treated in this model? As a negative dry deposition flux?

Authors: We have added 'emission of tracers' to the list of the processes considered in FLEXPART. More detailed information on how the emissions are treated in the model have been added to the second paragraph of section 2.8. We wrote: "For each data point of the observed sea to air flux, 10 000 air parcels were released from a 0.1˚ x 0.1˚ grid box at the ocean surface centered at the measurement location and loaded with the amount of the OVOCs prescribed by the observed emissions at this location."

Editor: Pp 7, line 30: "compared to a study along the South-East Florida coast" (or alternatively "in the coastal waters of South-East Florida"

Pp 8; line 16 "compared with"; check the whole document actually on this; according to me it is here "compared to" (compared with is used when things are similar, e.g., of magnitude) whereas "compared to" is used when you contrast things.

Pp 10: line 28 "for the entire ocean mixed layer" (to contrast this with the atmospheric mixed/boundary layer)

Authors: done

Editor: Pp 11, line 30: suggesting to change to; "For all measurement locations with a positive flux, reflecting the ocean being a source for atmospheric OVOCs," and would it be useful here to shortly indicate how many of all samples are indeed showing positive fluxes?

Authors: We changed it and added the number of samples used for the calculations.

Editor: Pp 12: line 30: "...driven by the hotspots east of 116°E which occur in all three OVOCs.." this statement reads weird: I would suggest to say, "..due to presence of areas east of 116 °E with large sources reflected in the measurements of all three OVOCs"

Pp 13: line 9-10: " hot spots exist at some short distance from the cruise track area. However, we think this is less likely as these hotspots"; hot spots or hotspots? And would it not be better to refer instead of a hotspot to " a strong source area"?

Authors: We changed it to "strong" or "large source areas".

Editor: Pp 13, line 35: list the reference Blitz et al. in the proper way

Authors: done

Editor: Pp 13, line 41: I happen to know the EMAC modelling system but not other readers; revise by or explaining the acronym or simply referring to EMAC as a "global chemistry-climate modelling system"

Authors: We added the information about EMAC. We wrote: "By combining their data with simulations from the global chemistry-climate modeling system (ECHMA/MESSy Atmospheric Chemistry, EMAC), Neumaier et al. showed …"